# Statistical characterization of erosion and sediment transport mechanics in shallow tidal environments. Part 2: suspended sediment dynamics

Davide Tognin[1,2], Andrea D'Alpaos[2], Luigi D'Alpaos[1], Andrea Rinaldo[1,3], and Luca Carniello[1]

[1]Department of Civil, Environmental, and Architectural Engineering, University of Padova, Padova, Italy
[2]Department of Geosciences, University of Padova, Padova, Italy
[3]Laboratory of Ecohydrology ECHO/IEE/ENAC, Ècole Polytechnique Fèdèrale de Lausanne, Lausanne, Switzerland
**Correspondence:** Davide Tognin (davide.tognin@unipd.it)

**Abstract.** A proper understanding of sediment resuspension and transport processes is key to the morphodynamics of shallow tidal environments. However, a complete spatial and temporal coverage of suspended sediment concentration (SSC) to describe these processes is hardly available, preventing the effective representation of depositional dynamics in long-term modelling approaches. Aiming to couple erosion and deposition dynamics in a unique synthetic theoretical framework, here we investigate SSC dynamics following a similar approach to that adopted for erosion (D'Alpaos et al., 2023). The analysis with the peak-over-threshold theory of SSC time series computed using a fully-coupled, bi-dimensional model allows us to identify interarrival times, intensities and durations of over-threshold events and test the hypothesis of modelling SSC dynamics as a Poisson process. The effects of morphological modifications on spatial and temporal SSC patterns are investigated in the Venice Lagoon, for which several historical configurations in the last four centuries are available. Our results show that, similarly to erosion events, SSC can be modelled as a marked Poisson process in the intertidal flats for all the analysed morphological lagoon configurations because exponentially distributed random variables well describe over-threshold events. Although erosion and resuspension are intimately intertwined, erosion alone does not suffice to describe also SSC because of the non-local dynamics due to advection and dispersion processes. The statistical characterization of SSC events completes the framework introduced for erosion mechanics and, together, they represent a promising tool to generate synthetic, yet realistic, time series of shear stress and SSC for the long-term modelling of tidal environments.

## 1 Introduction

Suspended sediment dynamics in shallow tidal systems play a significant role as they influence geomorphic and ecological processes, that ultimately determine the long-term morphodynamic evolution of coastal, estuarine and lagoonal landscapes (Woodroffe, 2002; Masselink et al., 2014). Physical processes that drive sediment resuspension and transport in tidal environments are influenced by different hydrodynamic and sedimentological factors over a wide range of spatial and temporal scales.

Both tide and waves represent key drivers controlling sediment entrainment and transport in shallow tidal environments (Wang, 2012). The tide rise and fall generate currents that propagate along the preferential pathways provided by the channel network (Hughes, 2012) but, as the tide overspills on the adjoining intertidal flats, it is strongly affected by shallower water and friction effects (Friedrichs and Madsen, 1992), so that its velocity and, hence, its resuspension capacity can diminish considerably. Whereas, wind waves with a typically short period can generate wave-orbital motions capable of resuspending intertidal-flat sediments (Anderson, 1972; Dyer et al., 2000; Carniello et al., 2005; Green, 2011). Therefore, stochastic wave-forced resuspension can increase locally, mainly under storm conditions, and can overcome the cyclic resuspension by tidal currents in generating high turbidity (Green et al., 1997; Ralston and Stacey, 2007; Sanford, 1994). Wave-driven resuspension and erosion together with tide- and wave-driven sediment transport give rise to mechanisms leading to basin-wide sediment movement, which strongly shape the morphology of shallow tidal systems (e.g., Nichols and Boon, 1994; Green and Coco, 2007; Carniello et al., 2011; Green and Coco, 2014). The repeated cycles of erosion, resuspension and deposition, that sediments may undergo, winnow fine particles from coarser ones and, thus, modify sediment distribution and textural properties of intertidal flats and subtidal platforms, influencing physical and biological processes (Dyer, 1989), light climate (Moore and Wetzel, 2000) and ecosystem productivity (Carr et al., 2010; Lawson et al., 2007; Carr et al., 2016; McSweeney et al., 2017).

Moreover, resuspension dynamics are mutually linked to numerous biological and ecological processes (Temmerman et al., 2007; Kirwan and Murray, 2007; D'Alpaos et al., 2007, 2011; Marani et al., 2013). Benthic vegetation and algae play a key role in increasing sediment stability of subtidal platforms (Nepf, 1999; Tambroni et al., 2016; Venier et al., 2014). In fact, the interaction of flexible vegetation and bedforms can reduce the effective bed shear stress and, consequently, sediment mobility. Similarly, the action of halophytic vegetation over salt marshes has a significant impact on landscape development, enhancing accretion, both by directly trapping inorganic sediment and by producing organic matter (Marani et al., 2013; D'Alpaos and Marani, 2016; Roner et al., 2016; Puppin et al., 2023). However, some studies have also suggested that, although vegetation anchors sediment through rooting and by slowing water flows, erosion and scour of the proximal sediments can also be enhanced (Temmerman et al., 2007; Tinoco and Coco, 2016). Microalgae, although small, may also heavily impact sediment erodibility. Indeed, extracellular polymeric secretions (EPS) of microphytobenthos can increase grain adhesion and consequently erosion threshold of the sedimentary substrate (Le Hir et al., 2007; Parsons et al., 2016; Chen et al., 2019). As a result, sediment resuspension decreases in the presence of EPS, which affects light availability and, in turn, microalgae proliferation, thus triggering positive feedback (Pivato et al., 2019). Benthic fauna can further modify the bed sediment by changing its geotechnical properties and erosion resistance (Widdows and Brinsley, 2002; Vu et al., 2017). Owing to the complexity of the underlying processes and the interplay between physical and biological drivers, sediment dynamics in shallow tidal systems are rather entangled.

Several numerical models have been developed to describe sediment transport and different techniques have been proposed to upscale the effects on the morphological evolution of tidal systems. For instance, explorative point-based models are extensively used to understand the relative importance of sediment transport processes, because of their simplified parametrization as well as their great conceptual value (Murray, 2007). Furthermore, their reduced computational burden is ideal to investigate trends over long-term time scales. For these reasons, point-based models have been largely adopted, for example, to examine salt-marsh fate under different sea level rise scenarios at the century time scale (D'Alpaos et al., 2011; Fagherazzi et al., 2012).

However, point-based models potentially miss spatial dynamics associated with sediment transport and, hence, might fail to represent interactions between different morphological units. More detailed, process-based models can fill this gap and account for sediment fluxes between different points up to the whole basin scale (e.g. Lesser et al., 2004; Carniello et al., 2012). But, because of the explicit description of the short-term interaction between hydrodynamics and sediment transport, the application of process-based models to the long-term time scale is often computationally expensive or even prohibitive. A widespread solution to overcome this limitation is to upscale the effects of short-term sediment transport on bed evolution by means of the so-called 'morphological factor', basically a multiplication factor to accelerate the computation of the effects on the morphology (Lesser et al., 2004; Roelvink, 2006).

These approaches implicitly assume that the morphological response of a system in the long term can be directly upscaled from the bed-level changes explicitly computed using a representative forcing condition on a much shorter time scale. However, as soon as the morphological evolution of a system is substantially affected by stochastic, episodic events, namely wind waves and storm surges (Tognin et al., 2021), and, therefore, cannot be represented as a continuous process (i.e. purely driven by the tide), this assumption may provide misleading results. Moreover, in tidal systems with fine sediments, because of the effect of consolidation, stratification and armouring of the sediment bed (Mehta et al., 1989), the morphological response is usually critically influenced by the magnitude and the time-history of events (Mathew and Winterwerp, 2022), which obviously cannot be reproduced by considering simplified, repetitive forcing conditions.

To explicitly model the effects of stochastic, morphologically-meaningful events as well as their temporal succession, a possible alternative would be to directly consider the physical processes responsible for the morphological evolution (i.e. erosion, transport and deposition of sediment) instead of upscaling the bed level changes. From this perspective, synthetic, statically-based models represent a particularly promising framework to reduce the computation burden associated with the explicit description of these processes through the use of independent Monte Carlo realizations. Notwithstanding the increasing popularity of statistically-based approaches for long-term modelling in hydrological and geomorphological sciences (e.g., Rodriguez-Iturbe et al., 1987; D'Odorico and Fagherazzi, 2003; Botter et al., 2013; Park et al., 2014), applications to tidal systems are still quite unusual (D'Alpaos et al., 2013; Carniello et al., 2016).

In order to explicitly describe sediment transport and bed evolution in a statistically-based framework, two different complementary processes need to be characterized: bottom shear stress (BSS), which can be considered a proxy for erosion, and suspended sediment concentration (SSC), which represents a measure of the sediment potentially available for deposition. To this goal, the characterization of BSS is provided by D'Alpaos et al. (2023). Here we aim to complete the proposed framework by statistically characterizing SSC and testing the possibility to describe suspended sediment dynamics as a Poisson process in long-term morphodynamic models.

SSC dynamics is usually characterized either by in situ point measurements (e.g., Wren et al., 2000; Gartner, 2004; Brand et al., 2020) or by remote sensing and satellite image analysis (Miller and McKee, 2004; Ruhl et al., 2001; Volpe et al., 2011). However, both these techniques have some drawbacks and do not offer the proper spatial and temporal coverage required for the statistical characterization. In situ measurements can provide an accurate description of the temporal dynamics of SSC, but lacks information on its spatial heterogeneity. Moreover, acoustic and optical sensors installed in point turbidity stations require

periodic cleaning to prevent failure due to biofouling. Whereas, satellite-based data can supply instantaneous information on SSC spatial variability, but are barely informative on its temporal dynamics. Indeed, SSC events can hardly be fully captured by satellites with fixed and often long revisit periods. Furthermore, intense SSC typically occurs during severe storms, frequently

characterized by clouds, which make satellite data useless. As a matter of fact, reliable long-term SSC time series at the basin scale, required for the statistical analysis performed herein, are seldom available. In order to overcome these shortcomings and to exploit measurements of in situ point observations and satellite images, these data can be combined to calibrate and test numerical models (Ouillon et al., 2004; Carniello et al., 2014; Maciel et al., 2021), thereby, using them as physically-based "interpolators" to compute temporal and spatial SSC dynamics required by this analysis. Here, we used a previously-calibrated

and widely-tested Wind Wave-Tidal Model (WWTM) (Carniello et al., 2005, 2011) coupled with a sediment transport model (Carniello et al., 2012) to investigate SSC dynamics.

      This study aims to verify if the proposed framework can be properly applied over long-term time scales and, hence, is independent of the specific morphological setting of a tidal basin. Hence, we perform the analysis on the Venice Lagoon, Italy (Figure 1), for which several historical morphological configurations are available in the last four centuries (Carniello et al.,

2009; D'Alpaos, 2010; Finotello et al., 2023). In particular, we considered the following six historical configurations: 1611, 1810, 1901, 1932, 1970, and 2012. For each of them, we run a one-year-long simulation forced with representative tidal and meteorological boundary conditions. The computed SSC time series have been analyzed on the basis of the peak-over-threshold (POT) theory, following the approach introduced by D'Alpaos et al. (2013) and expanding the analysis performed by Carniello et al. (2016) to study the statistics of SSC in the present configuration of the Venice Lagoon. Our analysis provides a spatial

and temporal characterization of resuspension events for the Venice Lagoon from the beginning of the seventeenth century to the present day, in order to show how morphological modifications affected sediment transport and to set up a stochastic framework to forecast future scenarios.

## 2    Materials and Methods

The Venice Lagoon (Figure 1) underwent different morphological changes over the last four centuries, mainly associated with

anthropogenic modifications (Carniello et al., 2009; D'Alpaos, 2010; Finotello et al., 2023). From the beginning of the fifteenth century, the main rivers (Brenta, Piave, and Sile) were gradually diverted in order to flow directly into the sea and prevent the lagoon from silting up, but this triggered the present-day sediment starvation condition. The inlets were provided with jetties between 1839 and 1934 and deep navigation channels were excavated to connect the inner harbour with the sea between 1925 and 1970 (D'Alpaos, 2010). The jetties deeply changed the hydrodynamics at the inlets establishing an asymmetric hydrody-

namic behaviour responsible for a net export of sediment toward the sea after their construction (Martini et al., 2004; Finotello et al., 2023), especially during severe storm events, which are responsible for the resuspension of large sediment volumes (Carniello et al., 2012). In general, these modifications, together with sea level rise, heavily influenced sediment transport triggering strong erosion processes in the following period. The net sediment loss clearly emerges from the comparison among the different surveys of the Venice Lagoon, which show a generalized deepening of tidal flats and subtidal platforms as well

as a reduction of salt-marsh area (Carniello et al., 2009). Indeed, in the last century, the average tidal-flat bottom elevation lowered from -0.51 m to -1.49 m above mean sea level (a.m.s.l.), while the salt-marsh area progressively shrank from 164.36 km$^2$ to 42.99 km$^2$ (Tommasini et al., 2019). This erosive trend displays a relative slowdown in the last 30 years because of the larger hydrodynamic forcing required to rework bed sediment at an increasing water depth (Finotello et al., 2023). However, repeated closures of storm-surge barriers designed to protect the city of Venice from flooding and known as Mo.S.E. system

are expected to further exacerbate this morphological degradation by cutting off significant supplies of inorganic sediments brought in by intense storm-surge events (Tognin et al., 2022). As a result, the morphological evolution of the lagoon in the last four centuries has been strongly affected by anthropogenic interventions, along with sea level rise.

To study the influence of these morphological changes on suspended sediment dynamics, we considered six different historical configurations of the Venice Lagoon, ranging from the beginning of the seventeenth century to today (Figure S1). The three

most ancient configurations (i.e. 1611, 1810, and 1901) were modelled by relying on historical maps, whereas the topographic surveys carried out by the Venice Water Authority (Magistrato alle Acque di Venezia) in 1932, 1970, and 2003 were used for the more recent ones (D'Alpaos, 2010; Finotello et al., 2023). Due to some morphological modifications at the three inlets associated with the Mo.S.E. system and almost completed in 2012, the 2003 configuration was updated, so we will refer to this configuration as the 2012 configuration. Each bathymetry and, hence, the elevation of grid elements refers to the local

mean sea level at the time when each survey was performed. For a detailed description of the methodology applied for the reconstruction of the historical configurations of the Venice Lagoon and additional information on the more recent bathymetric data, we refer the reader to Tommasini et al. (2019). Further details on the geomorphological setting and the implications on erosion and resuspension events are reported in D'Alpaos et al. (2023).

## 2.1 Numerical Model

The flow field and sediment transport in the six configurations of the Venice Lagoon are computed by using a numerical model, consisting of three modules. The coupling of the hydrodynamic module with the wind-wave module (WWTM) describes the hydrodynamic flow field together with the generation and propagation of wind waves (Carniello et al., 2005, 2011), while the sediment transport and the bed evolution module (STABEM) evaluates the sediment dynamics and the effects on the morphology (Carniello et al., 2012). All modules share the same computational grid.

The hydrodynamic module solves the 2-D shallow water equations using a semi-implicit staggered finite element method based on Galerkin's approach (Defina, 2000). The equations are suitably rewritten in order to deal with flooding and drying processes in morphologically irregular domains. Moreover, the hydrodynamic module provides the flow field characteristic used by the wind-wave module to simulate the generation and propagation of wind waves.

The wind-wave module (Carniello et al., 2011) solves the wave action conservation equation parametrized using the zero-

order moment of the wave action spectrum in the frequency domain (Holthuijsen et al., 1989). The spatial and temporal patterns of wave period are computed using an empirical function relating the mean peak wave period to the local wind speed and water depth (Young and Verhagen, 1996; Breugem and Holthuijsen, 2007; Carniello et al., 2011).

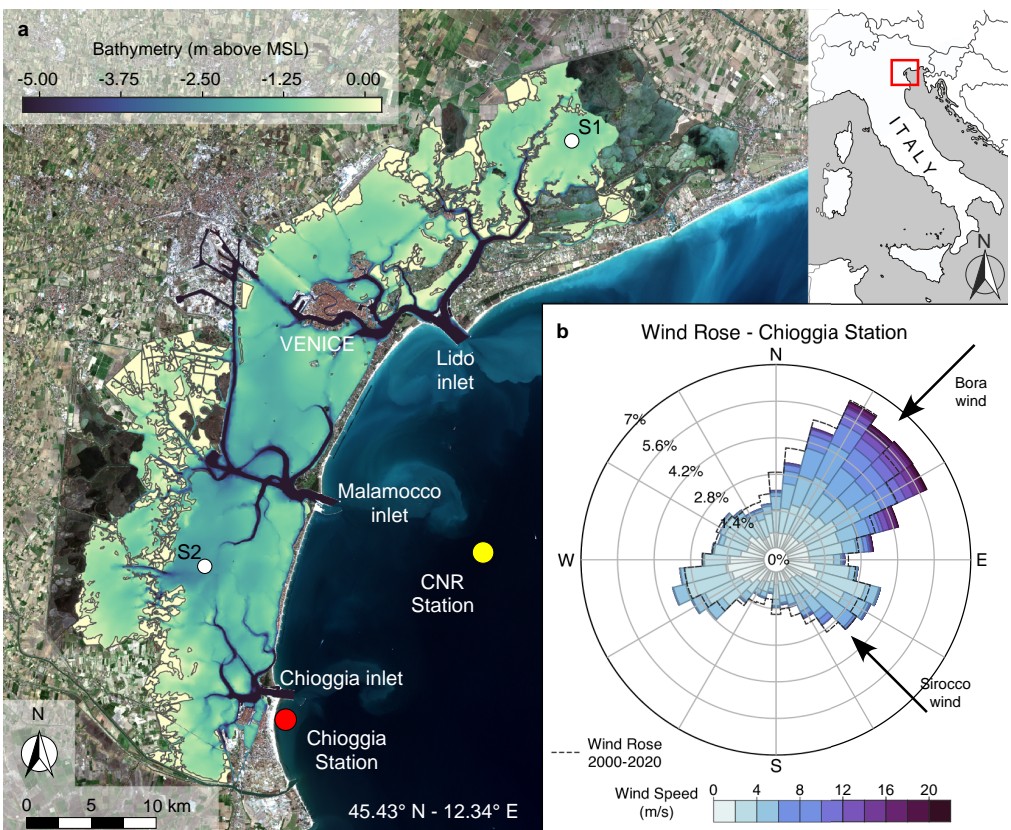

**Figure 1. Morphological features and wind conditions characterizing the Venice Lagoon. a**, Bathymetry of the Venice Lagoon (Base satellite image: Copernicus Sentinel data 2020, https://scihub.copernicus.eu/). The locations of the anemometric (Chioggia) and oceanographic (CNR Oceonographic Platform) stations are also shown, together with the locations of the two stations (S1 and S2) for which we provide detailed statistical characterization of over-threshold events. **b**, Wind rose for the data recorded at the Chioggia station in 2005. Dashed line shows the wind rose for the period 2000-2020.

The WWTM provides both current- and wave-induced bottom shear stresses. The bottom shear stress induced by currents, $\tau_{tc}$, is computed using the Strickler formulation, which, in the case of a turbulent flow over a rough wall, reads (Defina, 2000)

$$\tau_{tc} = \rho g Y \left( \frac{|\boldsymbol{q}|}{K_s^2 H^{10/3}} \right) \boldsymbol{q} \tag{1}$$

where $\rho$ is water density, $g$ is the gravity acceleration, $Y$ is the effective water depth (i.e. the actual volume of water per unit area), $\boldsymbol{q}$ is the flow rate per unit width, $K_s$ is the Strickler roughness coefficient, and $H$ is an equivalent water depth accounting for ground irregularities (Defina, 2000). The bottom shear stress induced by wind waves, $\tau_{ww}$, is computed as a function of the total horizontal orbital velocity at the bottom, $u_m$, and the wave friction factor, $f_w$, as follows

$$\tau_{ww} = \frac{1}{2} \rho f_w u_m^2 \tag{2}$$

The bottom orbital velocity, $u_m$, is evaluated by applying the linear theory and is also used, together with the wave period and median grain size, to compute the wave friction factor (Soulsby, 1997). Because of the non-linear interaction between the wave and current boundary layers, the total bottom shear stress, $\tau_{wc}$, is enhanced beyond the linear addition of the current- and wave-driven stresses. To account for this process, in the WWTM the empirical formulation suggested by Soulsby (1995, 1997) is adopted:

$$\tau_{wc} = \tau_{tc} + \tau_{ww} \left[ 1 + 1.2 \left( \frac{\tau_{ww}}{\tau_{ww} + \tau_{tc}} \right) \right] \tag{3}$$

The sediment transport and bed evolution module (STABEM, Carniello et al., 2012) is based on the solution of the advection-diffusion equation and Exner's equation:

$$\frac{\partial C_i Y}{\partial t} + \nabla \cdot (\boldsymbol{q} C_i) - \nabla \cdot (\boldsymbol{D_h} \nabla C_i) = E_i - D_i \qquad i = s, m \tag{4}$$

$$(1 - n) \frac{\partial z_b}{\partial t} = \sum_i (D_i - E_i) \tag{5}$$

where $C$ is the depth-averaged sediment concentration, $\boldsymbol{D_h}(x, y, t)$ represents the space- and time-dependent 2-D diffusion tensor, $E$ and $D$ are the entrainment and deposition rate of bed sediment, $z_b$ is the bed elevation and $n$ is the bed porosity, assumed equal to 0.4. The subscript $i$ refers to the sediment classes, that in shallow tidal environments are typically represented by non-cohesive (sand - $s$) and cohesive (mud - $m$) sediment. The relative local content of mud ($p_m$) can be used to mark off the transition between the cohesive or non-cohesive nature of the mixture and determines the critical value of the bottom shear stress. To discriminate between non-cohesive and cohesive behaviours, the threshold value of mud content $p_{mc}$ is set equal to 10 % (van Ledden et al., 2004).

The deposition rate of pure sand, $D_s$, is given by

$$D_s = w_s r_0 C_s \tag{6}$$

where $w_s$ is the sand settling velocity and $r_0$ is the ratio of near-bed to depth-averaged concentration, which is assumed constant and equal to 1.4 (Parker et al., 1987).

The deposition rate of pure cohesive mud, $D_m$, is computed using Krone's formula:

$$D_m = w_m C_m \max\{0; 1 - \tau_{wc}/\tau_c\} \tag{7}$$

where $w_m$ is the mud settling velocity, $\tau_{wc}$ is the bottom shear stress, and $\tau_d$ is the critical shear stress for deposition. The settling velocities, $w_s$ and $w_m$, are computed using the formulation proposed by van Rijn (1984) for solitary particles in clear and still water, thus not incorporating flocculation effects that are negligible for particle diameters larger than 20 $\mu$m (Mehta et al., 1989). The critical shear stress for deposition, $\tau_d$, largely varies among different tidal systems and, for the Venice Lagoon, we set $\tau_d = 1$ Pa on the basis of field measurements (Amos et al., 2004).

Both sand and mud erosion rates strongly depend on the cohesive nature of the mixture. The erosion rate for pure sand, $E_s$, is described by the van Rijn (1984) formulation when the mixture is non-cohesive ($p_m \leq p_{mc}$) and by the Partheniades'

formula for cohesive mixtures ($p_m > p_{mc}$):

$$E_s = \begin{cases} (1-p_m)w_s \cdot 1.5 \left( \dfrac{D_{50}/Y}{D_*^{0.3}} \right) T^{1.5} & \text{for } p_m \leq p_{mc} \\[2ex] (1-p_m) \cdot M_c T & \text{for } p_m > p_{mc} \end{cases} \tag{8}$$

The erosion rate for pure mud, $E_m$, is described by the formulation proposed by van Ledden et al. (2004) for non-cohesive mixtures ($p_m \leq p_{mc}$) and by the Partheniades' formula for cohesive mixtures ($p_m > p_{mc}$):

$$E_m = \begin{cases} \dfrac{p_m}{1-p_m} \cdot M_{nc} T & \text{for } p_m \leq p_{mc} \\[2ex] p_m \cdot M_c T & \text{for } p_m > p_{mc} \end{cases} \tag{9}$$

In Eqs. 8 and 9, $D_*$ denotes the dimensionless grain size and it is computed as $D_* = D_{50}[(s-1)g/\nu^2]^{1/3}$, where $s$ is the sediment-specific density and $\nu$ is the water kinematic viscosity; $T$ is the transport parameter; $M_{nc}$ and $M_c$ are the specific entrainments for non-cohesive and cohesive mixtures, respectively, which can be computed as (van Rijn, 1984; van Ledden et al., 2004):

$$M_{nc} = \alpha \frac{\sqrt{(s-1)gD_{50}}}{D_*^{0.9}}$$
$$M_c = \left( \frac{M_{nc}}{M_m} \cdot \frac{1}{1-p_{mc}} \right)^{\frac{1-p_m}{1-p_{mc}}} \cdot M_m \tag{10}$$

where $M_m$ is the specific entrainment for pure mud and it is set equal to $5 \cdot 10^{-2}$ g m s$^{-1}$ and the parameter $\alpha$ is equal to $1 \cdot 10^{-5}$ (Carniello et al., 2012).

The transport parameter, $T$, is usually defined as $T = \max\{0; \tau_{wc}/\tau_c - 1\}$ where $\tau_c$ is the critical shear stress for erosion and can be assumed to vary monotonically between the critical value for pure sand, $\tau_{cs}$, and the critical value for pure mud, $\tau_{cm}$, depending on the mud content (van Ledden et al., 2004):

$$\tau_c = \begin{cases} (1+p_m)\tau_{cs} & \text{for } p_m \leq p_{mc} \\[2ex] \dfrac{\tau_{cs}(1+p_{mc}) - \tau_{cm}}{1-p_{mc}}(1-p_m) + \tau_{cm} & \text{for } p_m > p_{mc} \end{cases} \tag{11}$$

However, this classic definition of the transport parameter describes a sharp transition between $T = 0$ and $T = \tau_{wc}/\tau_c - 1$ that does not take into account the spatial and temporal variability of both $\tau_{wc}$ and $\tau_c$. Indeed, in real tidal systems, the bottom shear stress slightly varies owing to the non-uniform flow velocity, wave characteristics and small-scale bottom heterogeneity, while the critical shear stress is also affected by the random grain exposure and bed composition in time and space. Hence, following the stochastic approach suggested by Grass (1970), both the total bottom shear stress, $\tau_{wc}$, and the critical shear stress for erosion, $\tau_c$, are treated as random variables ($\tau'_{wc}$, and $\tau'_c$, respectively) with lognormal distributions, and their expected values are those calculated by WWTM and STABEM. Consequently, the erosion rate depends on the probability that $\tau'_{wc}$ exceeds $\tau'_c$ (Carniello et al., 2012). The result of this stochastic approach is a smooth transition between $T = 0$ and $T = \tau_{wc}/\tau_c - 1$. The comparison with SSC field measurements shows a much better agreement of the stochastic approach compared to that of the

classic formulation (Supplementary information and Figure S3). Finally, erosion and deposition rates of sand and mud result in a variation of bed level and composition through time, which is computed using Eq. 5 and updating the local mud content.

The model has been widely calibrated and tested in the most recent configuration of the Venice Lagoon, i.e., when field data are available. Since the hydrodynamic model performance has been reported when considering the erosion dynamics (D'Alpaos et al., 2023), here we summarize the ability of the sediment transport model to reproduce SSC by reporting the standard Nash-Sutcliffe Model Efficiency (NSE) parameter computed when field data are available and refer the interested reader to the Supplementary Information (Figures S4 and S5) and the literature (Carniello et al., 2012; Tognin et al., 2022) for a more detailed comparison. Adopting the classification proposed by Allen et al. (2007), the model performance can be rated from excellent to poor (i.e., NSE > 0.65 excellent; 0.5 < NSE < 0.65 very good; 0.2 < NSE < 0.5 good; NSE < 0.2 poor). The STABEM model is very good to excellent in reproducing SSC ($NSE_{mean} = 0.65$, $NSE_{median} = 0.62$, $NSE_{std} = 0.17$, statistics are derived from calibration reported in Carniello et al. (2012), their Tables 2 and 3, and Tognin et al. (2022), their Table S2). Importantly, the sediment transport model not only correctly reproduces the magnitude of the SSC but also captures its modulation induced by tidal currents and wind-wave variations (Figures S4 and S5).

The coupled hydrodynamic and sediment transport models were used to perform one-year-long simulations within the six different computational grids representing the historical configurations of the Venice Lagoon and the portion of the Adriatic Sea in front of it. Hourly tidal level gauged at the Consiglio Nazionale delle Ricerche (CNR) Oceanographic Platform, located in the Adriatic Sea offshore of the lagoon, and wind velocities and directions recorded at the Chioggia anemometric station are imposed as boundary conditions (Figure 1).

All configurations were forced with tidal levels and wind climate measured during the whole year 2005,because the cumulative distribution frequency of wind velocity measured in 2005 is the closest to that of the whole period 2000-2020 (D'Alpaos et al., 2023), and, consequently, can be considered a representative year for the wind climate (Figure 1). Forcing all the historical configurations of the Venice Lagoon with the same wind and tidal conditions enables us to isolate the effects of the morphological modifications on the wind-wave field, hydrodynamics and sediment dynamics. Because bed elevation in each computational grid refers to the mean sea level at the time of each survey, we implicitly take into account the effects of historical relative sea level variations.

To correctly model SSC as well as bed evolution, the knowledge of the bed sediment composition is crucial. Sufficiently detailed, spatially-distributed grain-size data are available for the present-day configuration of the Venice Lagoon (Amos et al., 2004; Umgiesser et al., 2006). Using this dataset, Carniello et al. (2012) empirically related the median grain size $D_{50}$ to the local bottom elevation and the distance from the inlets:

$$D_{hf} = \begin{cases} \max\{300;\ 50(-h_f - 0.8)^{0.75}\} & \text{if } h_f \leq 1 \text{ m a.m.s.l.} \\ 15 & \text{if } h_f > 1 \text{ m a.m.s.l.} \end{cases} \tag{12}$$

$$D_{50} = D_{hf} + 100e^{-0.0097L^3} \tag{13}$$

where $h_f$ is the bottom elevation in m a.m.s.l.; $L$ is the linear distance from the closer inlet in km; $D_{50}$ and $D_{hf}$ are the grain diameter $\mu$m. This relationship describes a coarsening of the sediment grain size distribution at deeper locations (i.e. channels) and at shorter distances from the sea (Figure S2). Because bottom elevation and the distance from the inlet are the two main parameters describing the spatial variation in sediment grain size, we assume that this relationship holds independently on the specific morphological configuration of the Venice Lagoon and we used Eqs. 12 and 13 to compute the distribution of median grain size $D_{50}$ in all the six selected historical configurations.

The spatial distribution of mud content, $p_m$, is then computed as a combination of the local $D_{50}$ and the typical grain size of mud and sand fractions (Umgiesser et al., 2006)

$$p_m = 1 - \frac{\ln(D_{50}/D_m)}{\ln(D_s/D_m)} \tag{14}$$

where $D_m$ and $D_s$ are the typical grain size of mud and sand, respectively. Analysing the grain size distribution measured in the Venice Lagoon (Amos et al., 2004; Umgiesser et al., 2006), we set $D_m = 20$ $\mu$m and $D_s = 200$ $\mu$m.

Another important issue to consider when studying SSC dynamics in shallow tidal environments is the presence of benthic and halophytic vegetation, which both shelters the bed against the hydrodynamic action and increases the local critical shear stress for erosion because of the presence of roots. While the presence of halophytic vegetation over salt marshes is almost ubiquitous, reconstructing the presence of benthic vegetation on the tidal flats is much more difficult even for the present configuration of the lagoon and practically impossible for the ancient configurations (Goodwin et al., 2023). For the above reasons and for the sake of homogeneity, the simulations of the present study neglect the presence of benthic vegetation on the tidal flat and assume all salt-marsh platforms to be completely vegetated in each configuration of the lagoon, thus neglecting sediment resuspension over them (Christiansen et al., 2000; Temmerman et al., 2005).

## 2.2 Peak Over Threshold Analysis of SSC

The interplay among the different drivers that control suspended sediment dynamics in shallow tidal environments can be fully framed only by taking into account also its stochastic components, associated with wind waves and storm surges, which are largely responsible for the morphodynamic evolution of these systems (Carniello et al., 2011; Tognin et al., 2021). To this aim, in the present work, we statistically characterize the spatial and temporal dynamics of resuspension events by applying the peak-over-threshold theory (POT) (Balkema and de Haan, 1974) to the one-year-long time series of SSC computed with the numerical model described above for the different configurations of the Venice Lagoon.

Before applying the POT analysis, the SSC time series provided by the numerical simulations were low-pass filtered by applying a moving average procedure with a time window of 6 hours, in order to preserve the tide-induced modulation of the signal but, at the same time, to remove artificial upcrossing and downcrossing of the threshold, generated by short-term fluctuations. This pre-processing procedure prevents the identification of a false dependence of subsequent over-threshold events due to spurious fluctuations.

Once a proper threshold, $C_0$, is chosen, the POT identifies three different random variables: interarrival times, durations and intensities of the exceedances of the threshold. The interarrival time is defined as the time interval between two consecutive

upcrossings of the threshold, the duration of the events is the time elapsed between any upcrossing and the subsequent down-crossing of the threshold, and, finally, the intensity is calculated as the largest exceedance of the threshold in the time-lapse between an upcrossing and the subsequent downcrossing. These random variables are characterized by their probability density functions and the corresponding moments for any location in all the considered configurations of the Venice Lagoon, in order to provide a complete description of the SSC pattern. Besides synthetically characterising over-threshold events, these three variables can be combined to compute more complex metrics to describe SSC dynamics (e.g. the volume of sediment reworked in a selected time frame).

The nature of the stochastic processes can be determined by the analysis of the interarrival times distribution. Indeed, resuspension events can be mathematically modelled as a Poisson process if the interarrival times between subsequent exceedances of the threshold, $C_0$, are independent and exponentially distributed random variables (Cramér and Leadbetter, 1967; Gallager, 2013). Moreover, the memorylessness of the Poisson process guarantees that the number of events observed in disjoint sub-periods is an independent, Poisson-distributed random variable (Gallager, 2013). When the sequence of random events that define a 1-D Poisson process along the time axis can be associated with a vector of random marks that defines the duration and intensity of each over-threshold event, the process can be defined as a marked Poisson process. The distribution of these marks does not affect the chance to model the process as Poissonian, which, indeed, relies only on the exponentiality of interarrival times. However, when also duration and intensity are exponentially distributed, the set-up of a stochastic framework can be further simplified. In order to assess that over-threshold SSC events can be modelled as a marked Poisson process, we performed the Kolmogorov-Smirnov (KS) goodness of fit test on the distribution of the interarrival times, intensity and duration of over-threshold events.

In the POT analysis, the threshold value plays a critical role and its choice deserves careful attention. In the case of erosion dynamics (D'Alpaos et al., 2023), the identification of the threshold with the critical shear stress for erosion seems to be quite straightforward and has the advantage of preserving also the physical meaning of the process. Instead, when dealing with SSC, the absence of a clear physical threshold mechanism may make the identification of the threshold value less direct. The present analysis aims to characterize the bulk effect of morphologically meaningful SSC events, rather than to describe only the extreme events, and, simultaneously, to remove the weak resuspension events induced by periodic tidal currents that can be described as a recurrent, deterministic process. From this point of view, the choice of a threshold value, $C_0$, that identifies morphologically significant over-threshold SSC events, has to consider two opposite requirements. On the one hand, stochastic sediment concentration generated by storm-induced wind waves can be distinguished from tide-modulated daily concentration only if $C_0$ is large enough. On the other hand, too high values of $C_0$ either require a long, computationally prohibitive simulated time series or can lead to a non-informative analysis because of the large number of events unaccounted for. These observations narrow the range in which the threshold can be selected. The lower boundary is set by the SSC observed in the absence of wind and, therefore, associated exclusively with the tide. While the upper boundary has to be maintained well below the maximum observed values to consider all the morphologically meaningful events. In the specific case of the Venice Lagoon, to satisfy these requirements, the $C_0$ value has to fall between 30 and 60 mg l$^{-1}$, as suggested by in-situ SSC measurements (Carniello et al., 2012, 2014).

The sensitivity analysis performed on the present-day configuration of the Venice Lagoon (Carniello et al., 2016) suggests that the chance to model SSC events as a Poisson process is weakly affected by the specific threshold value in the above range. Indeed, using threshold values equal to 30, 40, 50 or 60 mg l$^{-1}$ hardly changes the areas where interarrival times are not exponentially distributed and, therefore, wind-induced SSC cannot be described as a Poisson process (Figure S6). On the basis of these observations and to allow the comparison among the different configurations, in the present analysis, we used a

constant threshold, $C_0$, equal to 40 mg l$^{-1}$.

## 3   Results and Discussion

We analyzed the time series of computed total SSC at any node of the computational grids reproducing the six selected configurations of the Venice Lagoon on the basis of the POT method, in order to characterize the over-threshold events in terms of interarrival times, peak excess and duration. We then performed the KS test (significance level $\alpha = 0.05$) to compare

the distributions of interarrival times with an exponential distribution, to test the hypothesis to model SSC events as a Poisson process. The KS test is also applied to peak excess and duration of the over-threshold events to test if these marks can be described by exponential distributions as well. The chance to model SSC dynamics as a Poisson process relies only on the exponentiality of interarrival times and it is not affected by the specific distribution of peak excess and duration. However, the set up of the final stochastic framework is simplified when also the duration and peak excess follows an exponential distribution.

Therefore, in the spatial distribution of the KS test results (Figure 2), we can identify three different situations:

1. SSC events cannot be described as a Poisson process, i.e. the KS test is not satisfied for interarrival times, in the dark blue areas;

2. SSC events are indeed a marked Poisson process because interarrival times, peak excesses and durations satisfy the KS test, and, thus, are exponentially distributed random variables, in the red areas;

3. SSC events still are a marked Poisson process but at least one between intensity and duration does not satisfy the KS test, i.e. although interarrival times follow an exponential distribution, at least one between intensity and duration does not, in the yellow areas.

The spatial distribution of mean interarrival times (Figure 3), mean intensities of peak excesses (Figure 4), and mean durations (Figure 5) of over-threshold events are shown at any location within each of the six historical configurations where SSC

events can be modelled as a Poisson process (i.e., the KS test is verified for interarrival times at significance level $\alpha = 0.05$). The mean values of these three random variables are shown where at least interarrival times are exponentially distributed (i.e. yellow areas in Figure 2) because the SSC dynamics can be modelled as a Poisson process although the marks are described by a distribution different from the exponential one and mean peak excess and mean duration are nonetheless informative.

The area of the lagoon where over-thresholds SSC events cannot be modelled as Poisson processes are mostly represented

by salt marshes and tidal channels in all configurations (see dark blue areas in Figure 2), similarly to the results for bottom shear stress (BSS) (D'Alpaos et al., 2023). On salt-marsh areas, both BSS and SSC thresholds ($\tau_C$ and $C_0$ respectively) are

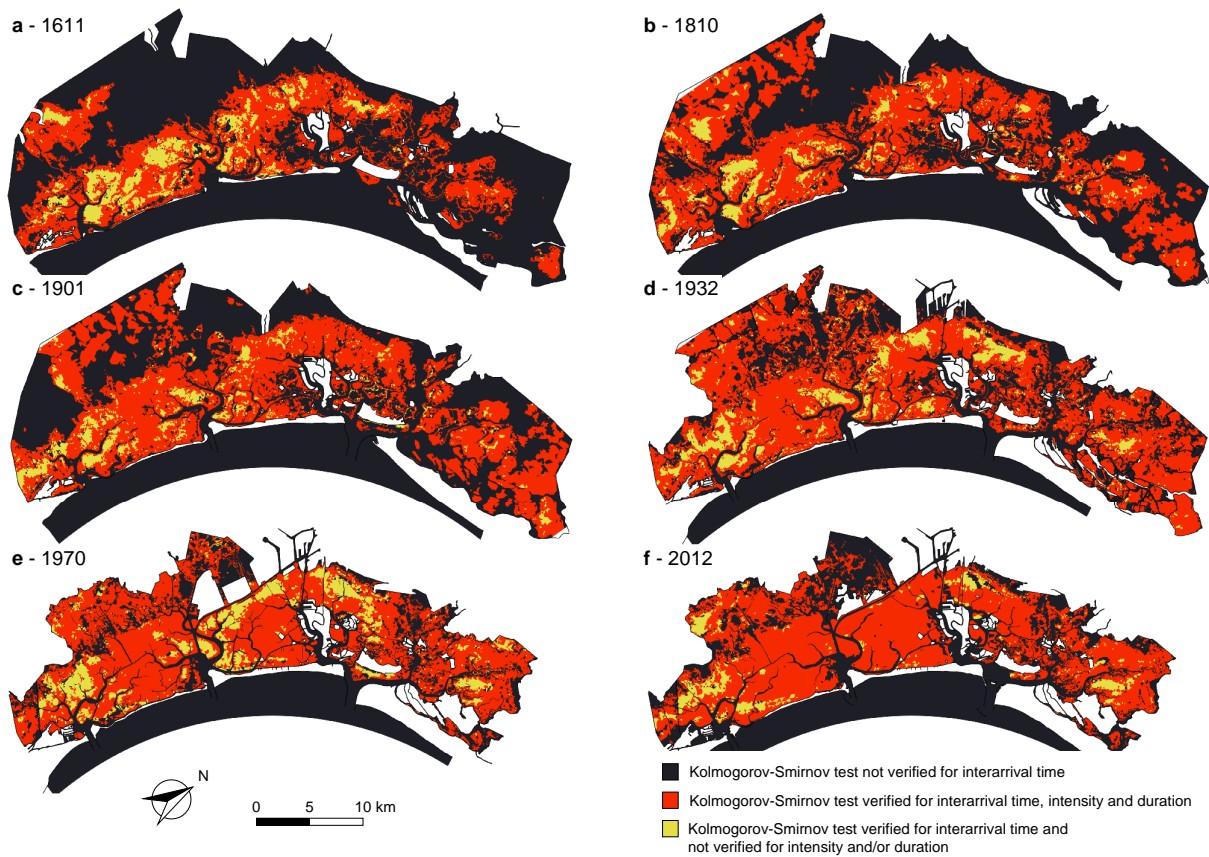

**Figure 2. Kolmogorov-Smirnov test for over-threshold SSC events.** Spatial distribution of Kolmogorov-Smirnov (KS) test at significance level ($\alpha = 0.05$) for the six different configurations of the Venice Lagoon: (**a**) 1611, (**b**) 1810, (**c**) 1901, (**d**) 1932, (**e**) 1970, and (**f**) 2012. In the maps we can distinguish areas where the KS test is: not verified (dark blue); verified for all the considered stochastic variables (interarrival time, intensity over the threshold and duration) (red); verified for the interarrival time and not for intensity and/or duration (yellow).

seldom exceeded (Figure S7 and S8), because the reduced water depth over the marsh prevents the propagation of large wind waves and the presence of halophytic vegetation limits sediment advection by promoting deposition and stabilizes the bottom preventing erosion (e.g., Möller et al., 1999; Temmerman et al., 2005; Carniello et al., 2005). Within the main tidal channels and at the three inlets, as happens for BSS, SSC dynamics are not Poissonian, but the reason why interarrival times of erosion and SSC events are not exponentially distributed are slightly different. In the main channel network and at the inlets, SSC exceeds the threshold value, $C_0$, very few times or it does not exceed the threshold at all, due to vertical dispersion mechanisms that decrease the local concentration of sediment in suspension in deeper areas (Figure S8). Conversely, BSS typically exceeds the threshold $\tau_c$ twice or four times a day (Figure S7) mainly because of the tide action but the BSS time evolution cannot be modelled as a Poisson process as confirmed by the KS test on interarrival times of over-threshold BSS events (D'Alpaos et al., 2023).

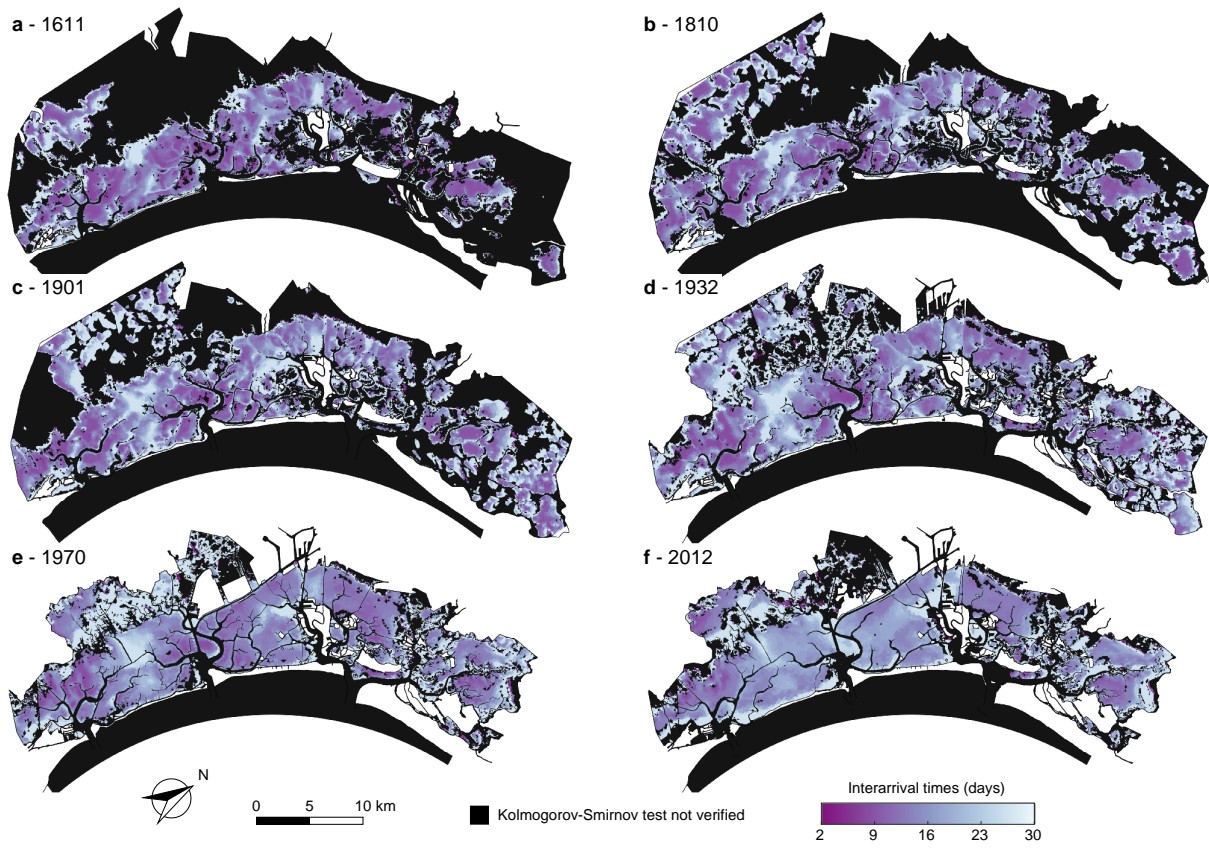

**Figure 3. Mean interarrival time of over-threshold SSC events.** Spatial distribution of mean interarrival times of over-threshold exceedances at sites where SSC events can be modelled as a marked Poisson process, as confirmed by the KS test ($\alpha = 0.05$) for the six different configurations of the Venice Lagoon: (**a**) 1611, (**b**) 1810, (**c**) 1901, (**d**) 1932, (**e**) 1970, and (**f**) 2012.

However, SSC events can be modelled as a Poisson process over wide areas of the six configurations of the Venice Lagoon, in particular over tidal flats and subtidal platforms (see red and yellow areas in Figure 2). As a consequence, SSC dynamics can be effectively modelled by using a synthetic framework based on Poisson processes over widespread portions of the different morphological configurations experienced by the Venice Lagoon in the last four centuries.

Large interarrival times (i.e., larger than 30 days, Figure 3) are observed on tidal flats close to the main channel network because dilution processes within higher water depth, enhanced by the higher velocities in these sites, reduce sediment concentration, and hence only severe, but infrequent, events can lead to an exceedance of the threshold. Sheltered areas are also characterized by large interarrival times as represented by the northern portion of the lagoon, which is protected by the mainland from the north-easterly Bora wind, which is the most intense and morphologically significant wind in the Venice Lagoon (Figure 1b), and where the presence of extensive salt-marsh areas continuously interrupts the propagation of wind waves. In this case, the reduced number of upcrossing events, and, consequently, large interarrival times is due to the sheltering action of

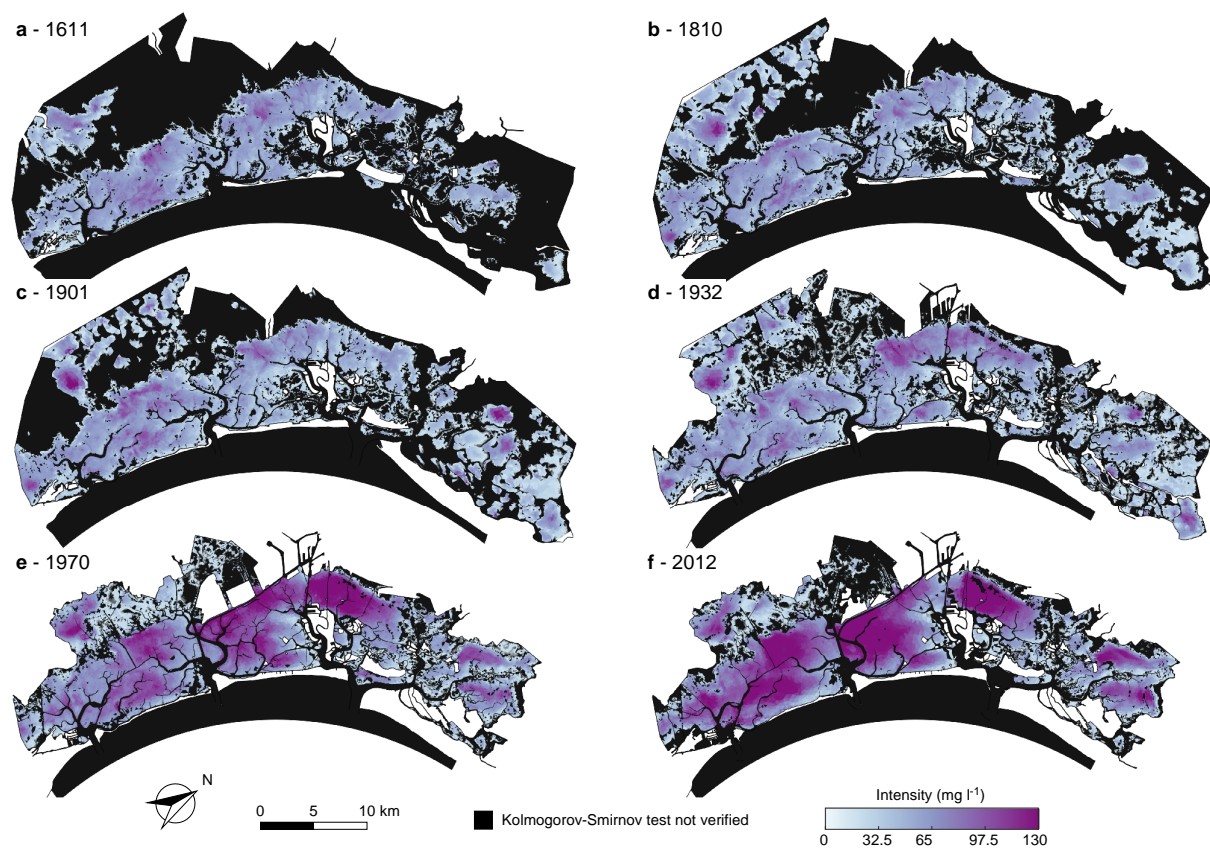

**Figure 4. Mean intensity of over-threshold SSC events.** Spatial distribution of mean intensity of peak excesses of over-threshold exceedances at sites where SSC events can be modelled as a marked Poisson process, as confirmed by the KS test ($\alpha = 0.05$) for the six different configurations of the Venice Lagoon: (**a**) 1611, (**b**) 1810, (**c**) 1901, (**d**) 1932, (**e**) 1970, and (**f**) 2012.

salt marshes and islands in reducing wind-wave resuspension. SSC events over the marsh platform slightly changed through centuries. In the three oldest configurations (i.e., 1611, 1810 and 1901) mainly because of the wide extent of salt marshes, re-
375 suspension events over salt marshes do not even reach the threshold, as shown by the number of upcrossing (Figure S8). In the more recent ones, where salt-marsh extent importantly decreases, marshes start experiencing some over-threshold SSC events because of the advection of sediment from the adjacent areas, but the lower number of upcrossing allows the mean interarrival time to assume large values.

Over wide tidal flat areas, where the threshold is exceeded in all the considered configurations, the mean interarrival time
generally slightly increases through the centuries (Figure S10a). This trend is more evident in the central and southern parts of the lagoon, where, because of the deepening experienced in the last century, the number of events able to resuspend sediments from the bottom decreased importantly, hence increasing the mean interarrival time of intense SSC events. In fact, over the central-southern shallow tidal flats of the four most ancient configurations, interarrival times present relatively low values (about

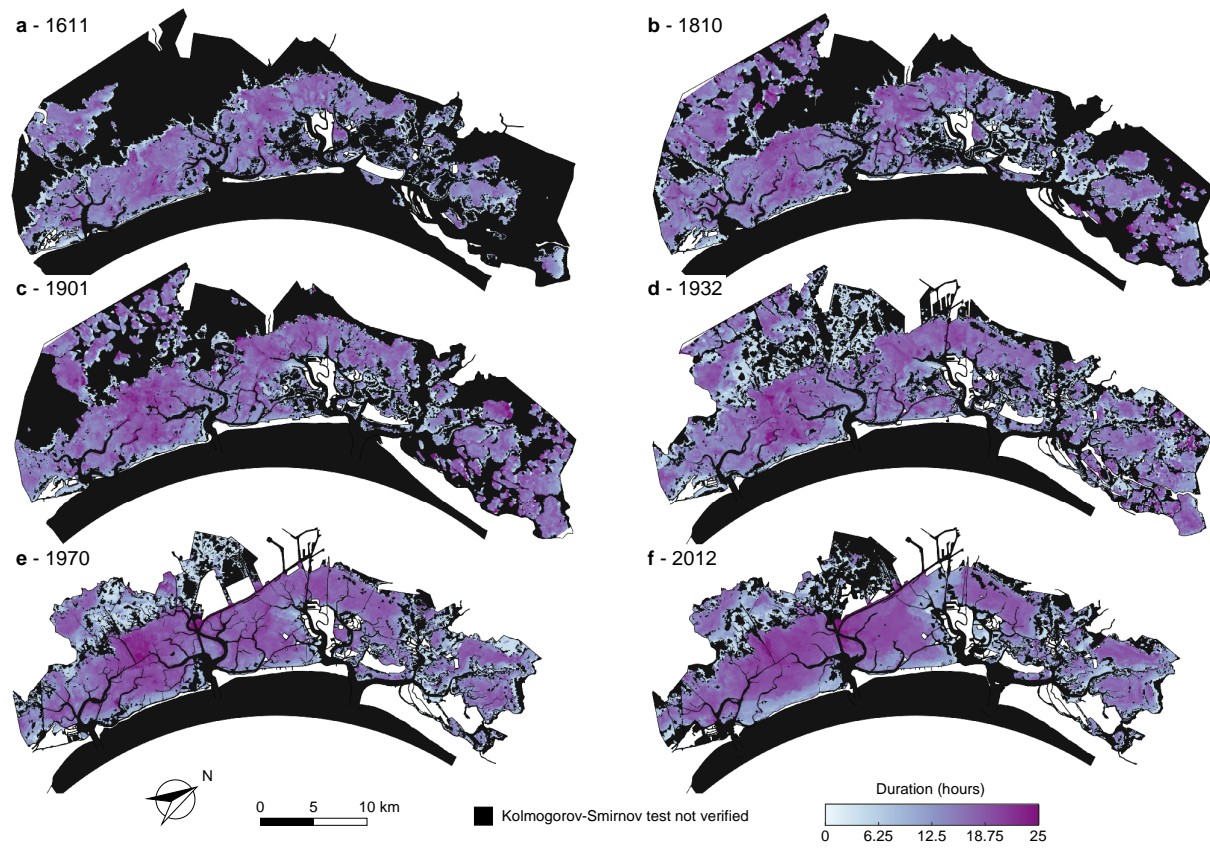

**Figure 5. Mean durations of over-threshold SSC events.** Spatial distribution of mean durations of over-threshold exceedances at sites where SSC events can be modelled as a marked Poisson process, as confirmed by the KS test ($\alpha = 0.05$) for the six different configurations of the Venice Lagoon: (**a**) 1611, (**b**) 1810, (**c**) 1901, (**d**) 1932, (**e**) 1970, and (**f**) 2012.

10 days), whereas they generally become longer (between 20 and 25 days) in the same areas in the more recent configurations.
On the contrary, in the better preserved, northern portion of the lagoon, where the fetch is continuously interrupted by islands, spits, and salt marshes also in the more recent configurations, the mean interarrival times experienced only slight changes over centuries. As an example, Figure 6a shows the mean interarrival times, $\lambda_t$, experienced by the "Palude Maggiore" tidal flat (station S1 in Figure 1) that do not vary remarkably over time. On the contrary, the subtidal flat at the watershed divide between the Chioggia and Malamocco inlets, known as "Fondo dei Sette Morti" (station S2 in Figure 1), display a much larger variation of $\lambda_t$ with decreasing interarrival times through centuries (Figure 6d). In the more ancient configurations, thanks to its relatively lower depth and its position sheltered by shallower tidal flats, station S2 experienced over-threshold events only during severe events. In the more recent configurations, over-threshold events become more frequent due to the deepening of the surrounding tidal flats, thus allowing larger waves and currents to propagate in this area and enhancing resuspension as well as suspended sediment transport.

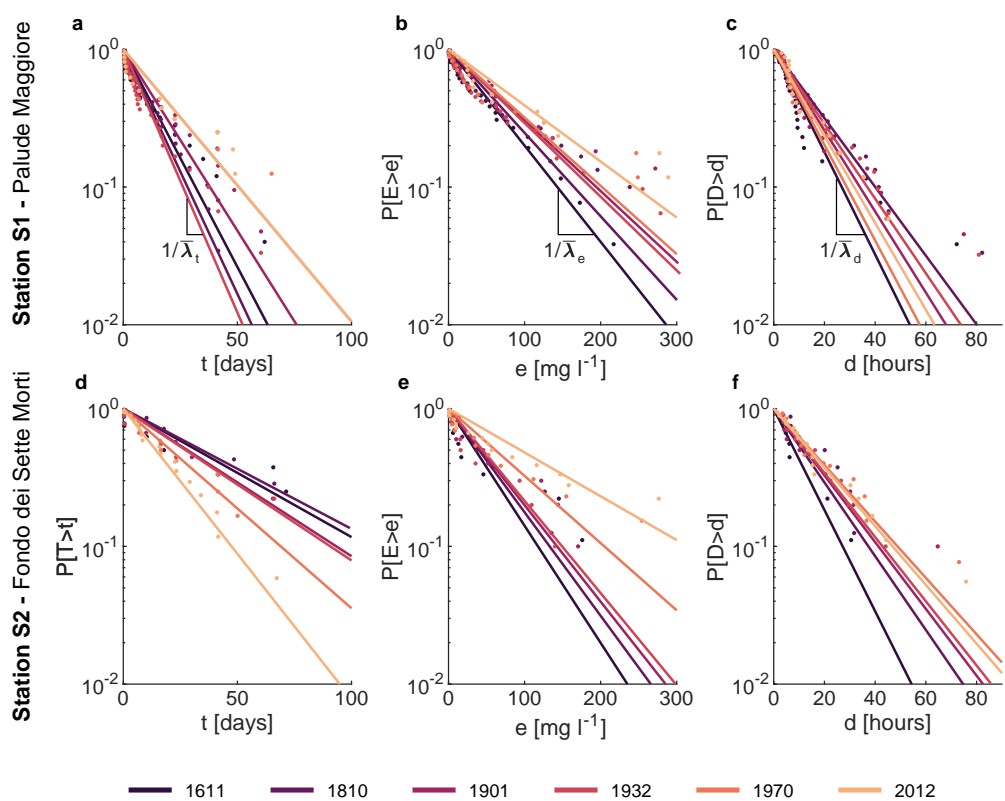

**Figure 6. Over-threshold SSC events at stations S1 and S2.** Statistical characterizations of over-threshold events at two stations S1 "Palude Maggiore" and S2 "Fondo dei Sette Morti" (see Figure 1a for locations) in the six configurations of the Venice Lagoon. Probability distributions of (**a-b**) interarrival times, $t$; (**c-d**) intensities of peak excesses of over-threshold exceedances, $e$; and (**e-f**) durations of over-threshold event, $d$. $\overline{\lambda}_t$ mean interarrival time, $\overline{\lambda}_e$ mean peak excess intensity, and $\overline{\lambda}_d$ mean duration.

The intensity of over-threshold events abruptly increases between 1932 and 1970 (Figure 4 and S10b). Indeed, SSC exceedance intensity maintains low mean values, generally below 60 mg l⁻¹, in all the configurations until 1932, thereafter it doubles on wide tidal-flat areas, especially in the central-southern lagoon and northwest of the city of Venice, where the action of wind waves is stronger because of the generalized deepening of those areas. This analysis confirms that the intensity increase is much more important in the central lagoon (station S2, Figure 6e) than in the northern part (station S1, Figure 6b).

Overall, over-threshold event durations slightly increase through the centuries (Figure 5 and S10c). However, two different trends can be distinguished in different portions of the lagoon, likewise interarrival times and intensities. The duration increase in the more pristine, northern portion of the basin is much lower than that in the central and southern lagoon due to the heavier morphological modifications the latter areas experienced (Figure 6c and f).

    SSC dynamics are affected by local entrainment and advection/dispersion processes from and toward the surrounding areas.

Furthermore, the local resuspension is highly influenced by the combined effect of tidal currents and wind waves, thus depend-

ing on current velocity, water depth, fetch, wind intensity and duration (Fagherazzi and Wiberg, 2009; Carniello et al., 2016). As a consequence, the mean values of the random variables characterizing SSC events present highly heterogeneous spatial patterns in the more ancient configurations of the Venice Lagoon due to their higher morphological complexity.

To describe the relationship between interarrival times, durations and intensities, the temporal cross-correlation between these three random variables was computed for each point within the six configurations of the Venice Lagoon (Figure S11, S12, S13). Duration of over-threshold exceedances and intensity of peak excesses are highly correlated in all the six considered configurations, suggesting that longer events are linked to more intense ones (Figure S11 and S14a). On the contrary, durations and interarrival times, as well as intensities and interarrival times display almost no correlation (Figure S11, S12 and S14b,c). These relations between interarrival time, intensity and duration back up the idea that, as for BSS dynamics (D'Alpaos et al., 2023), over-threshold SSC events can be modelled as a 3-D Poisson process in which the marks (intensity and duration of over-threshold events) are mutually dependent but independent on interarrival times.

As a result of the cause-effect relationship between the BSS (cause) and SSC (effect), their spatial and temporal dynamics show a high cross-correlation between interarrival times (Figure 7), intensity (Figure 8) and duration (Figure 9) of BSS and SSC over-threshold events. Recalling the absence of correlation between interarrival times and both intensities and durations for both BSS and SSC events, we can conclude that, when generating synthetic time series, interarrival times of BSS and SSC events are mutually dependent but not related to their intensity and duration. Intensities and durations of SSC are instead strongly correlated with the corresponding properties of BSS events.

Despite showing high similarity and correlation, BSS and SSC events are not identical. The BSS ultimately depends on the local hydrodynamics, i.e. the local value of the bed shear stress $\tau_{wc}$ produced by tidal currents and wind waves. On the contrary, the SSC is not only a function of the local entrainment but also of the suspended sediment flux from and towards the surrounding areas. As a result of the advection/dispersion processes, the spatial pattern of SSC is smoother than that of BSS.

The statistical characterization of over-threshold SSC events using their mean interarrival times, intensities and durations can be useful to estimate the total amount of reworked sediments. Although different portions of the lagoon experience different trends in these parameters depending on specific morphological modifications, a spatial average over the whole area where over-threshold SSC events can be described as Poisson processes shows that globally mean interarrival times and duration slightly vary and remain almost equal to about 30 days and 13 hours, respectively (Figure S10a and c). By contrast, intensity of the peak excess abruptly changes between 1932 and 1970. Between 1611 and 1932 the mean intensity maintains a value lower than 45 mg l$^{-1}$, but increases to 64 mg l$^{-1}$ in 1970 and further to 73 mg l$^{-1}$ in 2012 (Figure S10b).

This increase in the intensity of over-threshold SSC events, which is clearly associated with the generalized deepening of the tidal-flat areas, generates an increase in the amount of reworked sediment. This means that on average every month, for about 13 hours, the amount of sediment mobilized within the basin increases from about $2 \cdot 10^6$ kg in the three most ancient configurations to more than $6.8 \cdot 10^6$ kg in the 2012 configuration (Table 1). Besides directly boosting the amount of sediment available for export toward the open sea given the ebb-dominated character of the Venice Lagoon (Ferrarin et al., 2015; Finotello et al., 2023), the increase of suspended sediment also affects numerous biological and ecological processes that in turn influence the morphological evolution of the tidal system (e.g., Venier et al., 2014; Pivato et al., 2019).

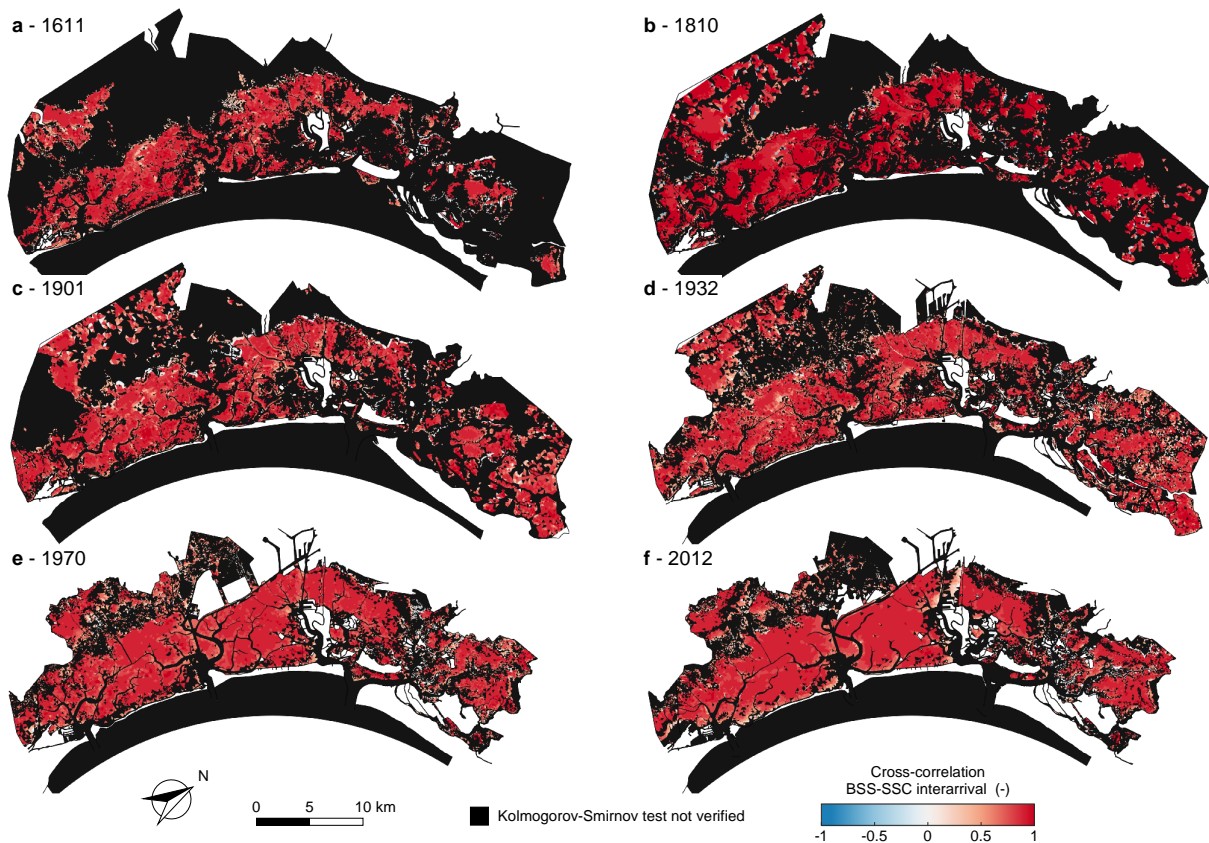

**Figure 7. Cross-correlation between interarrival times of over-threshold BSS and SSC events.** Spatial distribution of the cross-correlation between interarrival times of over-threshold BSS and SSC exceedances for the six different configurations of the Venice Lagoon: (**a**) 1611, (**b**) 1810, (**c**) 1901, (**d**) 1932, (**e**) 1970, and (**f**) 2012. Black identifies sites where over-threshold BSS or SSC events cannot be modelled as a marked Poisson process (i.e. the KS test is not verified for the interarrival time).

As already mentioned, modelling the morphodynamic evolution of tidal landscapes over long timescales (decades or centuries) necessarily requires the use of simplified approaches. However, a classical assumption of long-term evolution models is that the sediment supply is constant or monotonically related to mean water depth. The results presented in this study, together with those obtained for erosion events (D'Alpaos et al., 2023), demonstrate that the time series of both BSS and SSC can be described as marked Poisson processes with exponentially distributed interarrival times, intensities, and durations, thereby setting a framework for the synthetic generation of statistically significant external forcing factors (shear stress at the bottom and suspended sediment available in the water column) that should improve the reliability of long-term biomorphodynamic models with a limited increase in the number of parameters.

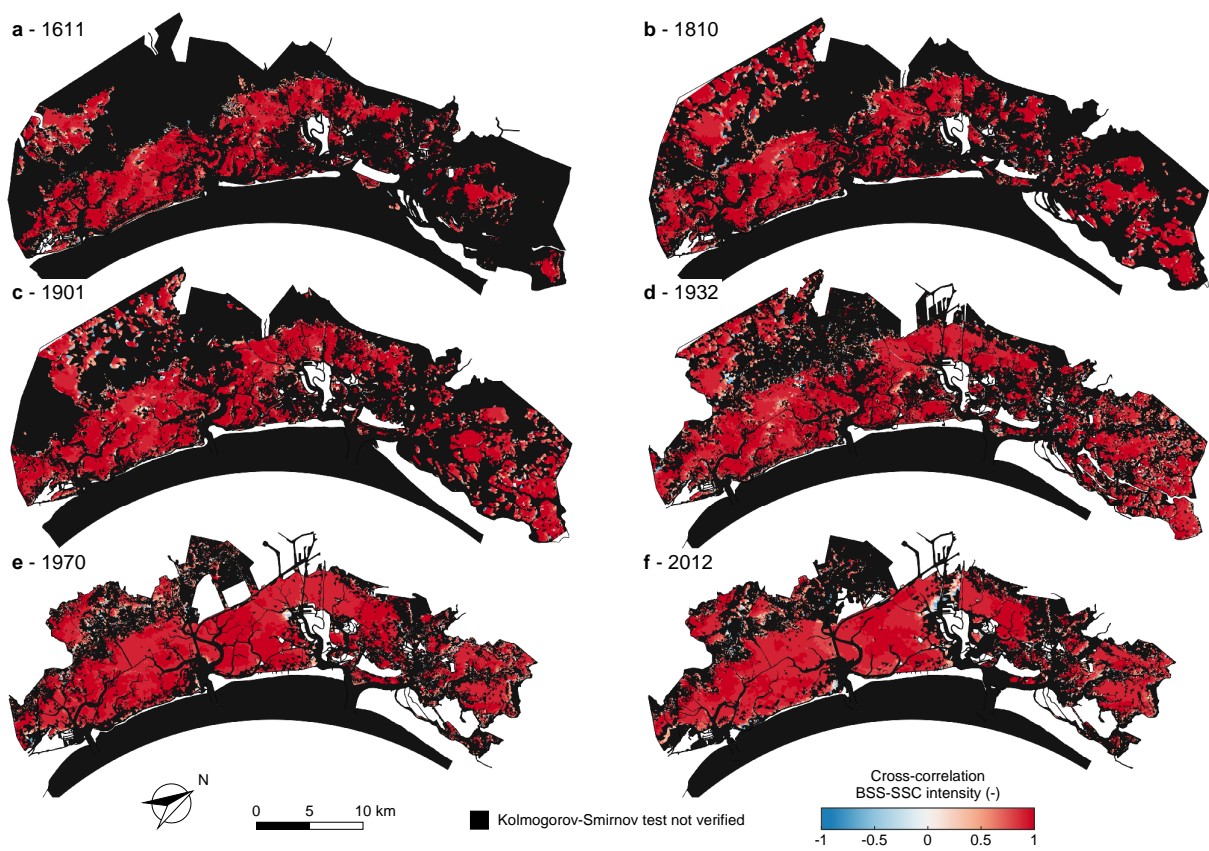

**Figure 8. Cross-correlation between intensities of over-threshold BSS and SSC events.** Spatial distribution of the cross-correlation between intensities of over-threshold exceedances BSS and SSC for the six different configurations of the Venice Lagoon: (**a**) 1611, (**b**) 1810, (**c**) 1901, (**d**) 1932, (**e**) 1970, and (**f**) 2012. Black identifies sites where over-threshold BSS or SSC events cannot be modelled as a marked Poisson process (i.e. the KS test is not verified for the interarrival time).

## 4 Conclusions

SSC dynamics in shallow tidal environments is usually investigated by means of field measurements or remote sensing analysis. However, due to the limited spatial coverage of field measurement and the temporal resolution of satellite images, long-term SSC dynamics at the basin scale are seldom available. Numerical models, once properly calibrated and tested, can provide reliable long SSC time series which can be used to statistically characterize the spatial and temporal variability of intense SSC events.

In the present study, we applied a custom-built, extensively tested, 2-D finite-element numerical model to reproduce SSC dynamics at basin scale in six historical configurations of the Venice Lagoon, covering a time span of four centuries. The computed SSC time series were analysed on the basis of the peak-over-threshold theory. Statistical analyses suggest that over-

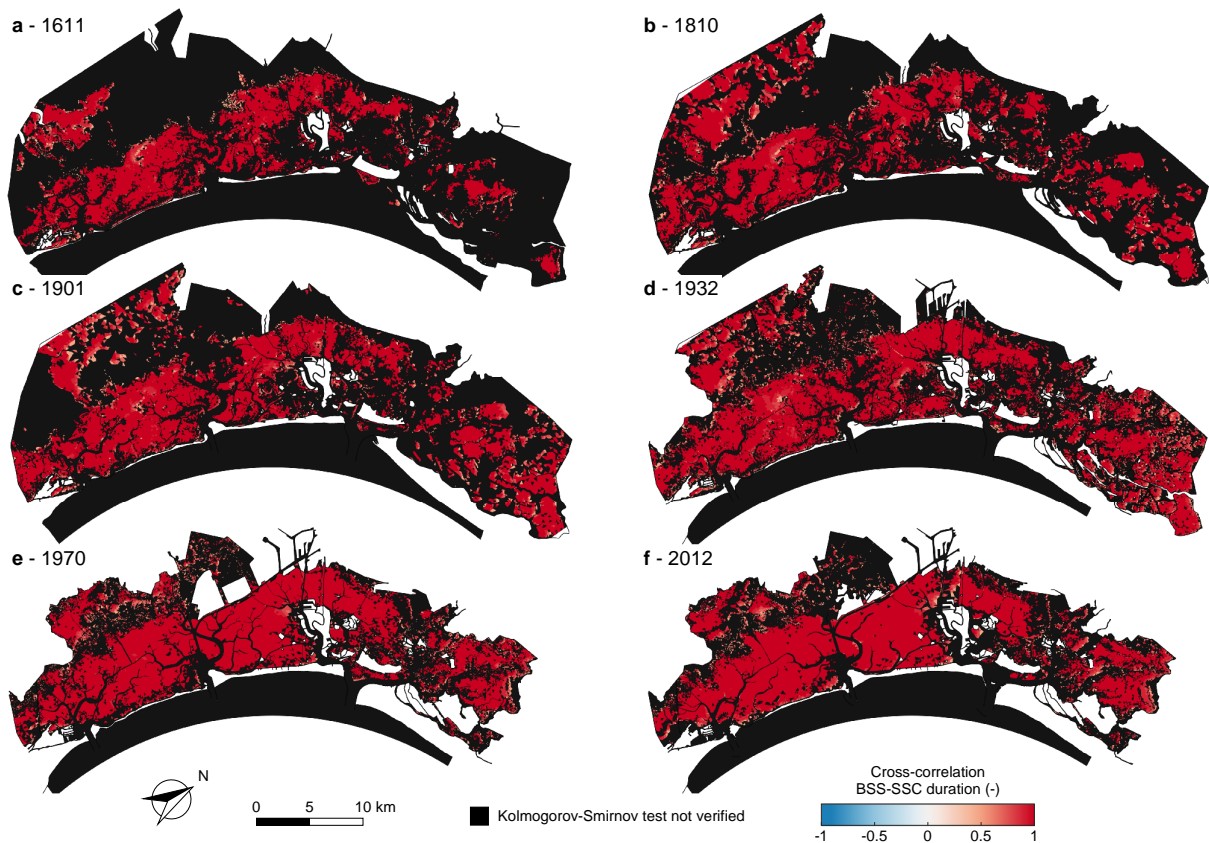

**Figure 9. Cross-correlation between durations of over-threshold BSS and SSC events.** Spatial distribution of the cross-correlation between durations of over-threshold exceedances BSS and SSC for the six different configurations of the Venice Lagoon: (**a**) 1611, (**b**) 1810, (**c**) 1901, (**d**) 1932, (**e**) 1970, and (**f**) 2012. Black identifies sites where over-threshold BSS or SSC events cannot be modelled as a marked Poisson process (i.e. the KS test is not verified for the interarrival time).

threshold SSC events can be modelled as a marked Poisson process over wide areas of all the selected configurations of the Venice Lagoon.

We found that, due to the morphological evolution experienced by the lagoon in the last four centuries, mean interarrival time, intensity and duration of over-threshold events generally increase through the centuries, generating slightly less frequent and longer, but stronger, resuspension events.

Furthermore, almost no correlation is shown to exist between durations and interarrival times of over-threshold exceedances and between intensities and interarrival times, whereas the intensity of peak excesses and duration are highly correlated. This

confirms that resuspension events can be modelled as a 3-D marked Poisson process with marks (intensity and duration) mutually dependent but independent on the interarrival times in all the historical configurations of the Venice Lagoon. Moreover, a comparison with the analysis of over-threshold BSS events shows that interarrival times, intensities and durations of both BSS

**Table 1. Sediment reworking in the historical configurations of the Venice Lagoon.** *area* (km$^2$): area of the lagoon where KS is verified; *h* (m): mean water depth of the area; $V_w$ (10$^6$ m$^3$): mean volume of water, obtained as product of area and water depth; *e* (mg l$^{-1}$): mean intensity of over-threshold SSC events; $S_{mob}$ (10$^6$ kg): sediment mobilized, assuming a triangular-shaped temporal evolution of over-threshold SSC events, with peak excess *e*.

| Year | area (km$^2$) | h (m) | $V_w$ (10$^6$ m$^3$) | $e$ (mg l$^{-1}$) | $S_{mob}$ (10$^6$ kg) |
|---|---|---|---|---|---|
| 1611 | 226.882 | 0.59 | 134.403 | 44.20 | 1.980 |
| 1810 | 294.649 | 0.43 | 127.022 | 40.84 | 1.729 |
| 1901 | 307.951 | 0.47 | 143.985 | 42.66 | 2.047 |
| 1932 | 350.166 | 0.54 | 188.661 | 43.49 | 2.734 |
| 1970 | 283.196 | 0.77 | 217.863 | 64.16 | 4.659 |
| 2012 | 270.022 | 1.04 | 279.969 | 73.21 | 6.832 |

and SSC events are mutually related but are complementary features because of the non-local dynamics due to advection and dispersion processes.

These findings, together with those obtained for BSS events (D'Alpaos et al., 2023), provide the basis to develop a theoretical framework for generating synthetic, yet statistically realistic, forcings to be used in the long-term morphodynamic modelling of shallow tidal environments, in general, and for the Venice Lagoon, in particular.

*Data availability.* All data presented in this study and used for the analysis of the suspended sediment concentration are available at https://researchdata.cab.unipd.it/id/eprint/729 (10.25430/researchdata.cab.unipd.it.00000729)

*Author contributions.* Conceptualization: Davide Tognin, Andrea D'Alpaos, Andrea Rinaldo, Luca Carniello;

Methodology: Davide Tognin, Andrea D'Alpaos, Luca Carniello;

Formal analysis and investigation: Davide Tognin;

Figures: Davide Tognin;

Writing - original draft preparation: Davide Tognin;

Writing - review and editing: all authors;

Funding acquisition: Andrea D'Alpaos, Luca Carniello;

Resources: Andrea D'Alpaos, Luca Carniello, Luigi D'Alpaos, Andrea Rinaldo;

Supervision: Andrea D'Alpaos, Luca Carniello.

*Competing interests.* The authors declare no competing interests.

*Acknowledgements.* TThis scientific activity was partially performed within the Research Programme Venezia2021, with contributions from the Provveditorato for the Public Works of Veneto, Trentino Alto Adige and Friuli Venezia Giulia, provided through the concessionary of State Consorzio Venezia Nuova and coordinated by CORILA, Research Line 3.2 (PI A.D.), the 2019 University of Padova project (BIRD199419) 'Tidal network ontogeny and evolution: a comprehensive approach based on laboratory experiments with ancillary numerical modelling and field measurements' (PI L.C.), the University of Padova SID2021 project, 'Unraveling Carbon Sequestration Potential by Salt-Marsh Ecosys-

tems' (P.I. A. D.), and the iNEST (Interconnected Nord-Est Innovation Ecosystem) project and received funding from the European Union Next-GenerationEU (National Recovery and Resilience Plan – NRRP, Mission 4, Component 2, – D.D. 1058 23/6/2022, ECS00000043 - CUP: C43C22000340006).

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
