# Peer review of "Statistical characterization of erosion and sediment transport mechanics in shallow tidal environments. Part 2: suspended sediment dynamics"

_EGUsphere, 2023_

## Referee Comment (RC2)

**Comments on "Statistical characterization of erosion and sediment transport mechanics in shallow tidal environments Part 2: suspended sediment dynamics"**

**1 Summary**

This work is part 2 of the study that introduces the idea of using random processes to model the wave-tidal-induced events along the coastal area. The Venice Lagoon, Italy is chosen as the study site due to the availability of multiple bathymetry surveys over the past centuries. The WWTM model coupled with STABEM sediment transport model is used to simulate morphodynamics. Statistics are extracted from simulation results. The author found that over-the-threshold suspended sediment concentration (SSC) events follows can be modeled as marked Poisson process. As Part 1 paper, this work paved a new way to upscale short-term simulations in a wave-tidal environment to long-term, while following the statistical characteristics. This paper has a very similar structure to the Part 1 paper, as well as employed identical analysis using morphodynamics results. However, different from hydrodynamics simulations, simulating the dynamics of SSC is much more complicated which means more uncertainties. Hence, I recommend the author show more validation of the modeling results. Maybe some changes also need to be adapted in the analysis to address the differences. Secondly, although I understand the idea behind the "threshold of SSC", in my opinion, the author did not show the physical meaning or mathematical definition of it, which makes it not well-defined, I recommend the author add more work to its definition. Thirdly, I believe when dealing with a time scale over 4 centuries, the climate can play an important role, and the analysis should take into consideration of it. As the revision in Part 1 paper, I recommend the author add more model details, and the choice of constants. There are more things that need to be addressed, which are listed below. Overall, I think this work has good potential, but still needs more work.

**2 Major comments**

1. As the first paper, the knowledge gap in this work is not clear from the literature reviews. The Poisson process is commonly used in describing the occurrence of events. The author needs to show that people have not used this technique in sediment transport events. For example, in Line 201, the author said that there are studies using Poisson processes to describe geophysical processes. Compare to those studies, the author should explicitly point out the new knowledge that the readers can gain from this work.

2. Line 137. Please introduce the equations that are used in the sediment transport model, particularly, the choice of entrainment relations for cohesive and non-cohesive materials, as well as the choice of parameters in the equations.

3. Line 144, the initial bed composition is very important in the simulation of suspended sediment concentration. More details are needed to show how the local bed composition is determined. Furthermore, it will be helpful to show data such as grain size distributions, and show how the sediment properties in this 2-sediment-class model are determined.

4. Line 151. The author mentioned that bottom shear stress and critical shear stress are determined using a stochastic approach from Grass (1970). Please give more details on how it is implemented. Is the implementation validated using benchmark tests? How do the results look like compared to the classic deterministic approach?

5. Line 182. In the first paper, the critical shear stress is a well-defined concept. While in Part 2, it is not clear what is the physical meaning of the "minimum-intensity threshold $C_0$" of SSC.

6. Line 201. The choice of $C_0 = 40$ mg/L needs to be justified.

7. Line 343. When talking about the comparison in the past centuries, it might be important to consider the impact of climate change.

8. In the introduction, the author stated that there exists field-measured data and remote sensing data for SSC. I think it is necessary to show the comparison between the simulated results and those datasets as model validation.

**3   Other comments**

1. In the captions in Figure 2, 3, 4, 5, 7, 8, 9 The description of subfigures is confusing. Recommend switching the order of the year and sub-figure numbering. For example, use "(a) 1611; (b) 1810; ..." instead.

---

## Author Comment (AC1)

**Author Response to Reviews of Earth Surface Dynamics Manuscript egusphere-2023-320**

**Statistical characterization of erosion and sediment transport mechanics in shallow tidal environments. Part 2: suspended sediment dynamics**

Davide Tognin[1,2], Andrea D'Alpaos[2], Luigi D'Alpaos[1], Andrea Rinaldo[1,3], and Luca Carniello[1]

[1] *Department of Civil, Environmental, and Architectural Engineering, University of Padova, Padova, Italy*
[2] *Department of Geosciences, University of Padova, Padova, Italy*
[3] *Laboratory of Ecohydrology ECHO/IEE/ENAC, Ècole Polytechnique Fèdèrale de Lausanne, Lausanne, Switzerland*
Correspondence: Davide Tognin (davide.tognin@unipd.it)

**Summary**

The authors wish to thank the Editorial Board and the Reviewers for their overall constructive and insightful comments on our paper, which significantly improved the manuscript and its readability.
We carefully revised the introduction following the Reviewers' suggestions in order to better highlight how the proposed approach contributes to filling the knowledge gap in long-term morphodynamic modelling and to better emphasize its complementarity with the companion paper on erosion dynamics. Moreover, we have significantly expanded the Method section, as suggested by the Reviewers. This expansion includes the description of equations implemented in the numerical sediment transport model, as well as an extended discussion on the choice of the threshold value to apply the peak-over-threshold analysis to suspended sediment concentration time series.
Finally, we provided additional details about some modelling choices that were not properly justified in the previous version of the manuscript, such as the selection of the boundary conditions and the initial bed sediment composition. Reviewers' suggestions on the companion paper that could have been applied also to this manuscript have been implemented, such as details on the study area, wind climate, choice of synthetic descriptors and model performance.
Overall, in the new version of the manuscript, we consistently revised the main text and importantly expanded the Supplementary Information, by adding the detailed model description and figures S2 to S6.
In the following, we discuss in detail all Reviewers' comments and show how we have addressed them in the revised manuscript, referencing line numbers in the revised manuscript with the track changes.
Please note that the Reviewers' comments are in blue, our detailed responses are in black, and the text of the revised manuscript is framed.

*Legend*

RC:     Reviewer Comment

AR:     Author Response

▢ :     Modified manuscript text

*Note*: References to reviewers' comments are indicated as RCx.x and numbered progressively.

**Reply to Reviewer #1**

RC1.1: This is an interesting paper combining a modeling approach and a statistical analysis of suspended sediment dynamics in the Venice Lagoon. The paper is well written, but the findings bring very little new insight, in comparison with its companion paper Part I on erosion dynamics. Although this is probably more of an editorial issue, I am questioning the relevance of making two papers out of this study. Indeed, both papers have basically the same structure, with very similar introduction and method sections. In addition, the results of both studies are highly correlated (see lines 300-305), which is not surprising as SSC dynamics (Part II) is itself highly correlated with erosion dynamics (Part I). To further support this, the authors keep referring to the companion paper on erosion dynamics to discuss their results on suspended sediment dynamics (section 3). … In conclusion, I don't deny the interest of this study, but I suggest to merge both companion papers into one.

AR: We thank the Reviewer for the overall positive and constructive comment on our manuscript. We must say that, while preparing the original version, we carefully examined the options of submitting one single manuscript or two companion papers. Let us better explain and justify the reasoning that led us to decide that the best option was represented by two companion papers.

As highlighted by the Reviewer, the objective of these two papers is to test the hypothesis and establish a theoretical framework for upscaling the effects of stochastic processes in the long-term morphodynamic modelling of shallow tidal environments. To this aim, both erosive and sediment transport dynamics obviously need to be taken into account and we deem that applying the same methodology to these two physical variables (namely bottom shear stress-BSS and suspended sediment concentration-SSC) makes undoubtedly the approach simpler, easier to be understood and more reasonable and justifiable. For this reason, the structure of these two manuscripts was intentionally kept similar.

Although the structure is similar and BSS and SSC are physically intertwined, the results are complementary and do not overlap. Indeed, we highlighted many differences between BSS and SSC dynamics, which surely deserve to be explained in detail. Following the Reviewer's suggestions, in the revised version, we further differentiate the two papers by rewriting the introductions to better highlight the complementarity of the two works (see our response to RC1.3 and RC2.1) and the Method sections to provide more details on the equations and the specific modelling choices (see our responses to RC1.2, 1.5, 1.6, and from RC2.2 to 2.6).

The choice of submitting two companion papers is also driven by the very practical reason of the manuscript length. A clear explanation of many details, which are necessary to understand the methodology we adopted, and a proper presentation of the analysis we performed to test the possibility to model both BSS and SSC dynamics as Poisson processes, require a quite long description and many visual elements (i.e. figures and tables). This is even clearer in the revised version of the two manuscripts, after the improvements suggested by the Reviewers. In total, the two revised manuscripts have 20 visual elements in the main text (10 for the BSS and 10 for the SSC) and 31 visual elements in the supplementary information (17 for the BSS and 14 for the SSC). Honestly, we deem that packing all this material into one single paper would jeopardize the readability of the manuscript because of the length and the need to continuously check the Supplementary information file, where too many visuals would necessarily be moved.

To conclude, we deem that the option of two companion papers offers the chance to clearly communicate our findings (compared to that of one single paper) and, at the same time, to highlight the strong link between our analysis of BSS and SSC dynamics (that may be missed with two separate papers in different journals). For all these reasons, we deem that merging these two papers into one would not be the optimal solution.

RC1.2: In the companion paper, the choice of a peak over threshold analysis is very natural, as erosion processes are physically triggered when the bed shear stress exceeds a threshold value. Here, the choice of such an analysis is less obvious, and determining an SSC threshold is highly arbitrary. Although the authors justify quite elegantly their choice of threshold value (line 212), they should at least discuss the sensitivity of their results and conclusions to this threshold value.

AR: We thank the Reviewer for his/her suggestion. We agree that the choice of the threshold value $C_0$ needs to be explained more in detail. To this aim, we have completely rewritten the subsection where we introduce the Peak-Over-Threshold analysis as follows:

(line 324) ~~Sediment transport dynamics in tidal environments are the results of the complex interplay among hydrodynamic, biologic, and geomorphologic processes. This interplay between different factors can be fully framed only by taking into account both its deterministic and stochastic components. As an example, Carniello et al. (2011) argued that morphological dynamics in the Venice Lagoon are mostly linked to a few severe resuspension events induced by wind waves, whose dynamics are markedly stochastic in the present configuration (D'Alpaos et al., 2013; Carniello et al., 2016). Measurements confirm that high SSC events are also important sediment suppliers for salt marshes (Tognin et al., 2021).~~

[revised manuscript text omitted]

~~According to the extreme value theory, a Poisson process emerges from a stochastic signal whenever enough high censoring threshold is chosen (Cramér and Leadbetter, 1967). However, as this present analysis is designed to remove only the weak resuspension events induced by periodic tidal currents, the critical threshold is well below the maximum observed values. As a consequence, the aim of the proposed analysis is to characterize the bulk effect of morphologically meaningful SSC events, rather than to describe the extreme events. Notwithstanding the increasing popularity of Poisson processes for the analytical modelling of the long-term evolution of geophysical processes controlled by stochastic drivers in hydrological and geomorphological sciences (e.g., Rodriguez-Iturbe et al., 1987; D'Odorico and Fagherazzi, 2003; Botter et al., 2013; Park et al., 2014; Bertassello et al., 2018), only in the last few years this approach has been adopted for tidal systems (D'Alpaos et al., 2013; Carniello et al., 2016) and the applications portray an encouraging framework.~~

For the Reviewer's convenience, we report here Figure S6 added to the Supplementary information showing the results of the KS test using different $C_0$ thresholds.

[Figure]

**Figure S6. Sensitivity analysis of the threshold $C_0$.** Spatial distribution of Kolmogorov-Smirnov (KS) test at significance level ($\alpha = 0.05$) for different values of the threshold, $C_0$: (a) 30 mg/l; (b) 40 mg/l; (c) 50 mg/l; (d) 60 mg/l. In the maps we can distinguish areas where the KS test is: not verified (dark blue); verified for all the considered stochastic variables (interarrival time, intensity over the threshold and duration) (dark red); verified for the interarrival time and not for intensity and/or duration (light red). Maps show little to no influence of the threshold value within the selected range on the possibility to model over-threshold SSC events as a Poisson process.

RC1.3: Something that intrigue me is hidden in lines 310-311. I am wondering if the results of this paper can be combined with the results of the companion paper to better constrained the erosion coefficient "e" (equation 3, companion paper). The value of this parameter is highly uncertain, given the values encountered in the literature range over more than one order of magnitude. If that is possible, that would be a very interesting result of this study.

AR: We appreciate the Reviewer's insightful observation on this point because the estimation of the erosion coefficient "e" is ideally one of the first validation steps that can be done by applying the statistically-based model we aim to develop, once the possibility to describe erosion and resuspension events as Poisson processes has been verified.

As we have now better clarified in the companion paper on BSS dynamics (see line 412 of the revised manuscript with the track changes), the calibration of the parameter "e" could not be performed solely on the basis of erosion dynamics because the erosion work represents the total potential erosion and thus completely disregards the possible settling of sediment carried in suspension, once the hydrodynamic conditions are favourable to deposition. The statistical characterization of SSC dynamics we derived in this paper aims to complete the framework proposed to describe BSS and, thus, to properly model the net bed evolution (i.e. calibrate also the parameter "e").

However, to correctly perform the analysis suggested by the Reviewer, a further step is still required: the set-up of the stochastic model to consider both erosion and resuspension dynamics. This may seem a very trivial and straightforward point, but, instead, it requires a careful and detailed explanation and validation, in which the point suggested by the Reviewer surely play a fundamental role. Even when the stochastic model will be available, directly comparing subsequent morphological configurations of the Venice Lagoon to "calibrate" the erosion coefficient "e" will be questionable, because the morphological evolution of the lagoon over the last century was deeply affected by human interventions (see our response to RC1.4 and RC2.7). Very likely the comparison between the result of the model and one of the lagoon configurations will highlight the effects of the anthropogenic interventions (i.e., excavation of large navigable channels, dredging, etc), which are not described by the statistically-based model.

In conclusion, we deem that the presentation of this model and related detailed analysis are beyond the aim of this study and cannot fit into one single paper, but we better highlighted the role of the SSC dynamics and its complementarity with erosion dynamics in the overall picture of the stochastic model we aim to develop, by modifying the text in several points as follows:
* * *
(line 1) *A proper understanding of sediment resuspension and transport processes* is key to the morphodynamics of shallow tidal environments. *However, a complete spatial and temporal coverage of suspended sediment concentration (SSC) to describe these processes is hardly available, preventing the effective representation of depositional dynamics in long-term modelling approaches.*  *Aiming to couple erosion and deposition dynamics in a unique synthetic theoretical framework, here we investigate SSC dynamics following a similar approach to that adopted for erosion (D'Alpaos et al., 2023).*

(line 23) *Although erosion and resuspension are intimately intertwined, erosion alone does not suffice to describe also SSC because of the non-local dynamics due*
* * *
*to advection and dispersion processes. The statistical characterization of SSC events completes the framework introduced for erosion mechanics and together they represent a promising tool to generate synthetic, yet realistic, time series of shear stress and SSC for the long-term modelling of tidal environments.*

(line 85) *To explicitly model the effects of stochastic, morphologically-meaningful events as well as their temporal succession, a possible alternative would be to directly consider the physical processes responsible for the morphological evolution (i.e. erosion, transport and deposition of sediment) instead of upscaling the bed level changes. From this perspective, a synthetic, statically-based model represents a particularly promising framework to reduce the computation burden associated with the explicit description of these processes through the use of independent Monte Carlo realizations. Notwithstanding the increasing popularity of statistically-based approaches for the long-term modelling in hydrological and geomorphological sciences (e.g., Rodriguez-Iturbe et al., 1987; D'Odorico and Fagherazzi, 2003; Botter et al., 2013; Park et al., 2014), applications to tidal systems are still quite unusual (D'Alpaos et al., 2013; Carniello et al., 2016).*
*In order to explicitly describe sediment transport and bed evolution in a statistically-based framework, two different complementary processes need to be characterized: bottom shear stress (BSS), which can be considered a proxy for erosion, and suspended sediment concentration (SSC), which represents a measure of the sediment potentially available for deposition. To this goal, the characterization of BSS is provided by D'Alpaos et al. (2023). Here we aim to complete the proposed framework by statistically characterizing SSC and testing the possibility to describe suspended sediment dynamics as a Poisson process in long-term morphodynamic models.*

(line 552) This confirms that resuspension events can be modelled as a 3-D marked Poisson process with marks (intensity and duration) mutually dependent but independent on the interarrival times in all the historical configurations of the Venice Lagoon. *Moreover, a comparison with the analysis of over-threshold BSS events shows that interarrival times, intensities and durations of both BSS and SSC events are mutually related but are complementary features because of the non-local dynamics due to advection and dispersion processes. These findings, together with those obtained for BSS events (D'Alpaos et al., 2023), provide the basis to develop a theoretical framework for generating synthetic, yet statistically realistic, forcings to be used in the long-term morphodynamic modelling of shallow tidal environments, in general, and for the Venice Lagoon, in particular.*

RC1.4: Lines 97-98: This somewhat contradicts results from the companion paper about erosion work, which increased then decreased over time, due to increase inter arrival time.

AR:    We think that this apparent contradiction may be due to the excessive conciseness in the description of the temporal succession of the morphological modification of the Venice Lagoon in the last century in the first version of the manuscript.

Indeed, after the strong erosion experienced between 1930 and 1970, the sediment loss displays a relative slowdown because it reached a plateau due to the more intense hydrodynamic forcing required to rework bed sediment at an increasing water depth, resulting from the erosion process (Carniello et al., 2009; D'Alpaos, 2010a; Finotello et al., 2023). From this point of view, this confirms and does not contradict our results about erosion work in the companion paper.

To avoid confusion, we modified the paragraph as follows:
* * *
(line 144) The Venice Lagoon (Figure 1) underwent different morphological changes over the last four centuries, in particular due to anthropogenic modifications (Carinello et al., 2009; D'Alpaos, 2010; Finotello et al., 2023). From the beginning of the fifteenth century, the main rivers (Brenta, Piave, and Sile) were gradually diverted in order to flow directly into the sea and prevent the lagoon from silting up, but this triggered *the present-day* sediment starvation condition.  The inlets were provided with jetties *between 1839 and 1934* and deep navigation channels were excavated to connect the inner harbour with the sea *between 1925 and 1970* (D'Alpaos, 2010). The jetties deeply changed the hydrodynamics at the inlets establishing an asymmetric hydrodynamic behaviour responsible for a net export of sediment toward the sea *after their construction* (Martini et al., 2004; Finotello et al., 2023), especially during severe storm events, which are responsible for the resuspension of large sediment volumes (Carniello et al., 2012). In general, these modifications, *together with sea level rise,* heavily influenced sediment transport triggering strong erosion processes *in the following period*.  The net sediment loss clearly emerges from the comparison among the different surveys of the Venice Lagoon, which show a generalized deepening of tidal flats and subtidal platforms as well as a reduction of salt-marsh area (Carniello et al., 2009). Indeed, in the last century, the average tidal-flat bottom elevation lowered from -0.51 m to -1.49 m above mean sea level (a.m.s.l.), while the salt-marsh area progressively shrank from 164.36 km$^2$ to 42.99 km$^2$ (Tommasini et al., 2019). *This erosive trend displays a relative slowdown in the last 30 years because of the larger hydrodynamic forcing required to rework bed sediment at an increasing water depth (Finotello et al., 2023).*
* * *
AR: We merged together our responses to RC1.5 and 1.6 because these two observations are closely linked. Following Reviewer's suggestions, we modified the text as follows:

(line 181) The hydrodynamic module solves the 2-D shallow water equations using a semi-implicit staggered finite element method based on Galerkin's approach (Defina, 2000). The equations are suitably rewritten in order to deal with flooding and drying processes in morphologically irregular domains.  *Moreover,*  the hydrodynamic module provides the flow field characteristic *used*  by the wind-wave module to simulate the generation and propagation of wind waves.

The wind-wave module (Carniello et al., 2011) solves the wave action conservation equation parametrized using the zero-order moment of the wave action spectrum in the frequency domain (Holthuijsen et al., 1989).  *The spatial and temporal patterns of wave period are computed using an empirical function relating the mean peak wave period to the local wind speed and water depth* (Young and Verhagen, 1996; Breugem and Holthuijsen, 2007; Carniello et al., 2011).

*The WWTM provides both current- and wave-induced bottom shear stresses. The bottom shear stress induced by currents, $\tau_{tc}$, is computed using the Strickler formulation, which, in the case of a turbulent flow over a rough wall, reads (Defina, 2000)*

$$\tau_{tc} = \rho g Y \left( \frac{|q|}{K_s^2 H^{10/3}} \right) q \tag{1}$$

*where $\rho$ is water density, $g$ is the gravity acceleration, $Y$ is the effective water depth (i.e. the actual volume of water per unit area), $\boldsymbol{q}$ is the flow rate per unit width, $K_s$ is the Strickler roughness coefficient, and $H$ is an equivalent water depth accounting for ground irregularities (Defina, 2000).*

*The bottom shear stress induced by wind waves, $\tau_{ww}$, is computed as a function of the total horizontal orbital velocity at the bottom, $u_m$, and the wave friction factor, $f_w$, as follows*

$$\tau_{ww} = \frac{1}{2} \rho f_w u_m^2 \tag{2}$$

*The bottom orbital velocity, $u_m$, is evaluated by applying the linear theory and is also used, together with the wave period and median grain size, to compute the wave friction factor (Soulsby, 1997). Because of the non-linear interaction between the wave and current boundary layers, the total bottom shear stress, $\tau_{wc}$, is enhanced beyond the linear addition of the current- and wave-driven stresses.*
*To account for this process, in the WWTM the empirical formulation suggested by Soulsby (1995, 1997) is adopted:*

$$\tau_{wc} = \tau_{tc} + \tau_{wc}\left[1 + 1.2\left(\frac{\tau_{ww}}{\tau_{ww} + \tau_{tc}}\right)\right] \qquad (3)$$

To further highlight the differences between the two companion papers, we reported in the Method section more details on the sediment transport model, which is exclusively used in the analysis of SSC and not for that of BSS. The modified version of the manuscript now reads:

[revised manuscript text omitted]

RC1.7: Line 153: Missing prime for the total BSS?

AR:     Thank you for noting. Added (see also modified text in the previous response).

---

## Author Comment (AC2)

**Author Response to Reviews of Earth Surface Dynamics Manuscript egusphere-2023-320**

**Statistical characterization of erosion and sediment transport mechanics in shallow tidal environments. Part 2: suspended sediment dynamics**

Davide Tognin[1,2], Andrea D'Alpaos[2], Luigi D'Alpaos[1], Andrea Rinaldo[1,3], and Luca Carniello[1]

[1] *Department of Civil, Environmental, and Architectural Engineering, University of Padova, Padova, Italy*
[2] *Department of Geosciences, University of Padova, Padova, Italy*
[3] *Laboratory of Ecohydrology ECHO/IEE/ENAC, Ècole Polytechnique Fèdèrale de Lausanne, Lausanne, Switzerland*
Correspondence: Davide Tognin (davide.tognin@unipd.it)

**Summary**

The authors wish to thank the Editorial Board and the Reviewers for their overall constructive and insightful comments on our paper, which significantly improved the manuscript and its readability.
We carefully revised the introduction following the Reviewers' suggestions in order to better highlight how the proposed approach contributes to filling the knowledge gap in long-term morphodynamic modelling and to better emphasize its complementarity with the companion paper on erosion dynamics. Moreover, we have significantly expanded the Method section, as suggested by the Reviewers. This expansion includes the description of equations implemented in the numerical sediment transport model, as well as an extended discussion on the choice of the threshold value to apply the peak-over-threshold analysis to suspended sediment concentration time series.
Finally, we provided additional details about some modelling choices that were not properly justified in the previous version of the manuscript, such as the selection of the boundary conditions and the initial bed sediment composition. Reviewers' suggestions on the companion paper that could have been applied also to this manuscript have been implemented, such as details on the study area, wind climate, choice of synthetic descriptors and model performance.
Overall, in the new version of the manuscript, we consistently revised the main text and importantly expanded the Supplementary Information, by adding the detailed model description and figures S2 to S6.
In the following, we discuss in detail all Reviewers' comments and show how we have addressed them in the revised manuscript, referencing line numbers in the revised manuscript with the track changes.
Please note that the Reviewers' comments are in blue, our detailed responses are in black, and the text of the revised manuscript is framed.

*Legend*

RC:    Reviewer Comment

AR:    Author Response

☐ :    Modified manuscript text

*Note*: References to reviewers' comments are indicated as RCx.x and numbered progressively.

**Reply to Reviewer #2**

RC2.0: This work is part 2 of the study that introduces the idea of using random processes to model the wave-tidal-induced events along the coastal area. The Venice Lagoon, Italy is chosen as the study site due to the availability of multiple bathymetry surveys over the past centuries. The WWTM model coupled with STABEM sediment transport model is used to simulate morphodynamics. Statistics are extracted from simulation results. The author found that over-the-threshold suspended sediment concentration (SSC) events follows can be modeled as marked Poisson process. As Part 1 paper, this work paved a new way to upscale short-term simulations in a wave-tidal environment to long-term, while following the statistical characteristics.

This paper has a very similar structure to the Part 1 paper, as well as employed identical analysis using morphodynamics results. However, different from hydrodynamics simulations, simulating the dynamics of SSC is much more complicated which means more uncertainties. Hence, I recommend the author show more validation of the modeling results. Maybe some changes also need to be adapted in the analysis to address the differences.

Secondly, although I understand the idea behind the "threshold of SSC", in my opinion, the author did not show the physical meaning or mathematical definition of it, which makes it not well-defined, I recommend the author add more work to its definition.

Thirdly, I believe when dealing with a time scale over 4 centuries, the climate can play an important role, and the analysis should take into consideration of it. As the revision in Part 1 paper, I recommend the author add more model details, and the choice of constants.

There are more things that need to be addressed, which are listed below. Overall, I think this work has good potential, but still needs more work.

AR: We thank the Reviewer for the overall positive assessment of our manuscript and for his/her constructive suggestions that contributed to improving the quality and clarity of our manuscript.

The three main points raised here are discussed in detail in the following. Concerning the model and its calibration, we presented the modification to the revised manuscript in our replies to RC2.2, 2.3, 2.4 and 2.8. Our responses to RC2.5 and 2.6 deal with the definition and choice of an SSC threshold. Finally, the Reviewer's concern about the time scale and the effects of climate are discussed in our response to RC2.7.

RC2.1: As the first paper, the knowledge gap in this work is not clear from the literature reviews. The Poisson process is commonly used in describing the occurrence of events. The author needs to show that people have not used this technique in sediment transport events. For example, in Line 201, the author said that there are studies using Poisson processes to describe geophysical processes. Compare to those studies, the author should explicitly point out the new knowledge that the readers can gain from this work.

AR: Following Reviewer's suggestion, we deeply revised the introduction as follows:

(line 63) *Several numerical models have been developed to describe sediment transport and different techniques have been proposed to upscale the effects on the morphological evolution of tidal systems. For instance, explorative point-based models are extensively used to understand the relative importance of sediment transport processes, because of their simplified parametrization as well as their great conceptual value (Murray, 2007). Furthermore, their reduced computational burden is ideal to investigate trends over long-term time scales. For these reasons, point-based models have been largely adopted, for example, to examine salt-marsh*

*fate under different sea level rise scenarios at the century time scale (D'Alpaos et al., 2011; Fagherazzi et al., 2012). However, point-based models potentially miss spatial dynamics associated with sediment transport and, hence, might fail to represent interactions between different morphological units. More detailed, process-based models can fill this gap and account for sediment fluxes between different points up to the whole basin scale (e.g. Lesser et al., 2004; Carniello et al., 2012). But, because of the explicit description of the short-term interaction between hydrodynamics and sediment transport, the application of process-based models to the long-term time scale is often computationally expensive or even prohibitive. A widespread solution to overcome this limitation is to upscale the effects of short-term sediment transport on bed evolution by means of the so-called 'morphological factor', basically a multiplication factor to accelerate the computation of the effects on the morphology (Lesser et al., 2004; Roelvink, 2006). These approaches implicitly assume that the morphological response of a system in the long term can be directly upscaled from the bed-level changes explicitly computed using a representative forcing condition on a much shorter time scale. However, as soon as the morphological evolution of a system is substantially affected by stochastic, episodic events, namely wind waves and storm surges (Tognin et al., 2021), and, therefore, cannot be represented as a continuous process (i.e. purely driven by the tide), this assumption may provide misleading results. Moreover, in tidal systems with fine sediments, because of the effect of consolidation, stratification and armouring of the sediment bed (Mehta et al., 1989), the morphological response is usually critically influenced by the magnitude and the time-history of events (Mathew and Winterwerp, 2022), which obviously cannot be reproduced by considering simplified, repetitive forcing conditions. To explicitly model the effects of stochastic, morphologically-meaningful events as well as their temporal succession, a possible alternative would be to directly consider the physical processes responsible for the morphological evolution (i.e. erosion, transport and deposition of sediment) instead of upscaling the bed level changes. From this perspective, synthetic, statically-based models represent a particularly promising framework to reduce the computation burden associated with the explicit description of these processes through the use of independent Monte Carlo realizations. Notwithstanding the increasing popularity of statistically-based approaches for long-term modelling in hydrological and geomorphological sciences (e.g., Rodriguez-Iturbe et al., 1987; D'Odorico and Fagherazzi, 2003; Botter et al., 2013; Park et al., 2014), applications to tidal systems are still quite unusual (D'Alpaos et al., 2013; Carniello et al., 2016).*

[revised manuscript text omitted]

~~the basin. The latter represents an interesting goal, being the use of stochastic frameworks particularly promising for long-term studies, as pointed out by their increasing popularity in hydrology and geomorphology to describe the long-term behaviour of geophysical processes (e.g., Rodriguez-Iturbe et al., 1987; D'Odorico and Fagherazzi, 2003; Botter et al., 2013; Park et al., 2014; Bertassello et al., 2018) Nonetheless, applications to tidal systems are still quite uncommon (D'Alpaos et al., 2013; Carniello et al., 2016).~~ *Our analysis provides a spatial and temporal characterization of resuspension events for the Venice Lagoon from the beginning of the seventeenth century to the present day, in order to show how morphological modifications affected sediment transport and to set up a stochastic framework to forecast future scenarios.*

To better highlight the overall picture of the stochastic model, the role of the statistical characterization of the SSC dynamics and its complementarity with erosion dynamics, we also modified the abstract as follows:

(line 1) A proper understanding of *sediment resuspension and transport processes* is key to the morphodynamics of shallow tidal environments. *However, a complete spatial and temporal coverage of suspended sediment concentration (SSC) to describe these processes is hardly available, preventing the effective representation of depositional dynamics in long-term modelling approaches.*  *Aiming to couple erosion and deposition dynamics in a unique synthetic theoretical framework, here we investigate SSC dynamics following a similar approach to that adopted for erosion (D'Alpaos et al., 2023).* ~~A complete spatial and temporal coverage of suspended sediment concentration (SSC) required to effectively characterize resuspension events is hardly available through observation alone, even combining point measurements and satellite images, but it can be retrieved by properly calibrated and tested numerical models. We analyzed one-year-long time series of SSC computed by a bi-dimensional, finite-element model in six historical configurations of the Venice Lagoon in the last four centuries. Following the peak-over-threshold theory, we statistically characterized suspended sediment dynamics by analyzing interarrival times, intensities and durations of over-threshold SSC events.~~ *The analysis with the peak-over-threshold theory of SSC time series computed using a fully-coupled, bi-dimensional model allows us to identify interarrival times, intensities and durations of over-threshold events and test the hypothesis of modelling SSC dynamics as a Poisson process. The effects of morphological modifications on spatial and temporal SSC patterns are investigated in the Venice Lagoon, for which several historical configurations in the last four centuries are available.* Our results show that, similarly to erosion events, SSC can be modelled as a marked Poisson process in the intertidal flats for all the *analysed morphological lagoon*  configurations  because exponentially distributed random variables well describe  over-threshold events. ~~Moreover, interarrival times, intensity and duration describing local erosion and over-threshold SSC events are highly related, although not identical because of the non-local dynamics of suspended sediment transport related to advection and dispersion processes. Owing to this statistical characterization of SSC events, it is possible to generate synthetic, yet realistic, time series of SSC for the long-term~~

*modelling of shallow tidal environments.*  *Although erosion and resuspension are intimately intertwined, erosion alone does not suffice to describe also SSC because of the non-local dynamics due to advection and dispersion processes. The statistical characterization of SSC events completes the framework introduced for erosion mechanics and, together, they represent a promising tool to generate synthetic, yet realistic, time series of shear stress and SSC for the long-term modelling of tidal environments.*

RC2.2: Line 137. Please introduce the equations that are used in the sediment transport model, particularly, the choice of entrainment relations for cohesive and non-cohesive materials, as well as the choice of parameters in the equations.

AR: As we noted in our responses to the revision of the companion paper, it is not an easy task to find a good compromise between conciseness and completeness in the description of already-published models.

Thanks to the Reviewers' comments, we realized that the summary we provided in the first version of the manuscript was lacking some necessary details, such as the equations used in the sediment transport model, which are particularly relevant for the subsequent analysis presented, although they cannot be considered the main focus of this paper. For this reason, we included in the Method section the equations implemented in the STABEM model to compute the sediment transport, by modifying the text as follows:

[revised manuscript text omitted]

The description of the transport parameter T (line 257) is reported in our response to RC2.4.

RC2.3: Line 144, the initial bed composition is very important in the simulation of suspended sediment concentration. More details are needed to show how the local bed composition is determined. Furthermore, it will be helpful to show data such as grain size distributions, and show how the sediment properties in this 2-sediment-class model are determined.

AR: To meet the Reviewer's suggestion, we added more details on the initial bed composition in the Method section as follows:

(line 297) *To correctly model SSC as well as bed evolution, the knowledge of the bed sediment composition is crucial. Sufficiently detailed, spatially-distributed grain-size data are available for the present-day configuration of the Venice Lagoon (Amos et al., 2004; Umgiesser et al., 2006). Using this dataset, Carniello et al. (2012) empirically related the median grain size $D_{50}$ to the local bottom elevation and the distance from the inlets:*

$$D_f = \begin{cases} \max\{300;\ 50(-h_f - 0.8)^{0.75}\} & \text{if } h_f \leq 1 \text{ m a.m.s.l.} \\ 15 & \text{if } h_f > 1 \text{ m a.m.s.l.} \end{cases} \qquad (12)$$

$$D_{50} = D_{hf} + 100e^{-0.0097L^3} \qquad (13)$$

*where $h_f$ is the bottom elevation in m a.m.s.l.; L is the linear distance from the closer inlet in km; $D_{50}$ and $D_{hf}$ are the grain diameter μm. This relationship describes a coarsening of the sediment grain size distribution at deeper locations (i.e. channels) and at shorter distances from the sea (Figure S2). Because bottom elevation and the distance from the inlet are the two main parameters describing the spatial variation in sediment grain size, we assume that this relationship holds independently on the specific morphological configuration of the Venice Lagoon and we used Eqs 12 and 13 to compute the distribution of median grain size $D_{50}$ in all the six selected historical configurations.*
*The spatial distribution of mud content, $p_m$, is then computed as a combination of the local $D_{50}$ and the typical grain size of mud and sand fractions (Umgiesser et al., 2006)*

$$p_m = 1 - \frac{\ln(D_{50}/D_m)}{\ln(D_s/D_m)} \qquad (14)$$

*where $D_m$ and $D_s$ are the typical grain size of mud and sand, respectively. Analysing the grain size distribution measured in the Venice Lagoon (Amos et al., 2004; Umgiesser et al., 2006), we set $D_m = 20\mu m$ and $D_s = 200\mu m$.*

RC2.4: Line 151. The author mentioned that bottom shear stress and critical shear stress are determined using a stochastic approach from Grass (1970). Please give more details on how it is implemented. Is the implementation validated using benchmark tests? How do the results look like compared to the classic deterministic approach?

AR: Following Reviewer's suggestion, we added more details on the stochastic approach for the computation of the transport parameter as follows:

(line 257) *The transport parameter, T, is usually defined as $T = \max\{0; \tau_{wc}/\tau_c - 1\}$ where $\tau_c$ is the critical shear stress for erosion and can be assumed to vary monotonically between the critical value for pure sand, $\tau_{cs}$, and the critical value for pure mud, $\tau_{cm}$, depending on the mud content (van Ledden et al, 2004):*

$$\tau_c = \begin{cases} (1 + p_m)\tau_{cs} & \text{for } p_m \leq p_{mc} \\ \frac{\tau_{cs}(1+p_{mc})-\tau_{cm}}{1-p_{mc}}(1 - p_m) + \tau_{cm} & \text{for } p_m > p_{mc} \end{cases} \qquad (11)$$

*However, this classic definition of the transport parameter describes a sharp transition between T = 0 and $T = \tau_{wc}/\tau_c - 1$ that does not take into account the*

*spatial and temporal variability of both $\tau_{wc}$ and $\tau_c$. Indeed, in real tidal systems, the bottom shear stress slightly varies owing to the non-uniform flow velocity, wave characteristics and small-scale bottom heterogeneity, while the critical shear stress is also affected by the random grain exposure and bed composition in time and space. Hence, following the stochastic approach suggested by Grass (1970), both the total bottom shear stress, $\tau_{wc}$, and the critical shear stress for erosion, $\tau_c$, are treated as random variables ($\tau_{wc}{}'$, and $\tau_c{}'$, respectively) with lognormal distributions, and their expected values are those calculated by WWTM and STABEM. Consequently, the erosion rate depends on the probability that $\tau_{wc}{}'$ exceeds $\tau_c{}'$ ( Carniello et al., 2012). The result of this stochastic approach is a smooth transition between $T = 0$ and $T = \tau_{wc}/\tau_c - 1$. The comparison with SSC field measurements shows a much better agreement of the stochastic approach compared to that of the classic formulation (Supplementary information and Figure S3).*

A detailed description of the implementation and validation of this approach was already provided by Carniello et al. (2012). We deem that repeating this information would be redundant and unnecessarily lengthens the manuscript compared to the little benefit for the reader. Therefore, we believe that these details would better fit the supplementary material. We report here for the Reviewer's convenience the text and the figure we added as Supplementary Information:

***Supplementary Information***

***The stochastic approach for the computation of the transport parameter***

*The transport parameter, T, is usually defined as*

$$T = \max\left\{0; \frac{\tau_{wc}}{\tau_c} - 1\right\} \qquad (1)$$

*where $\tau_{wc}$ is the total bottom shear stress and $\tau_c$ is the critical shear stress for erosion.*

*This definition describes a sharp transition between $T = 0$ and $T = \tau_{wc}/\tau_c - 1$ that cannot take into account the spatial and temporal variability of both $\tau_{wc}$ and $\tau_c$ in real tidal systems. Indeed, the bottom shear stress is very unsteady because of the non-uniform flow velocity, wave characteristics and small-scale bottom heterogeneity within a computational element, while the critical shear stress is also affected by the random grain exposure and bed composition in time and space.*

*Similarly to the stochastic approach proposed by Grass (1970), we assume that both bed shear stress, $\tau_{wc}$, and critical shear stress, $\tau_c$, are random and distributed according to a log-normal probability density function Grass, 1970; Bridge and Bennett, 1992). Therefore, we can write*

$$T = \frac{1}{\tau_c} \int_0^\infty \left[ \int_{\widetilde{\tau_c}}^\infty (\widetilde{\tau_{wc}} - \widetilde{\tau_c}) \cdot p_{wc}(\widetilde{\tau_{wc}}) d\widetilde{\tau_{wc}} \right] \cdot p_c(\widetilde{\tau_c}) d\widetilde{\tau_c} \qquad (2)$$

*where $p_{wc}(\cdot)$ and $p_c(\cdot)$ are the probability density function of $\tau_{wc}$ and $\tau_c$ respectively, and $\widetilde{\tau_{wc}}$, $\tau_{wc}$ are the correspondent dummy variables of integration. The result of this stochastic approach is a smooth transition between $T = 0$ and $T = \tau_{wc}/\tau_c - 1$.*

*An adequate interpolation of Eq. 2, which is implemented in the numerical model, is given by*

$$T = -1 + \left(1 + \left(\frac{\tau_b}{\tau_c}\right)^\varepsilon\right)^{1/\varepsilon} \qquad\qquad (3)$$

*where ε is a non-dimensional calibration parameter, that accounts for the shape of the log-normal distribution.*

*We report here as an example a comparison of the measured and computed SSC at the LT7 station (see Figure S2) for an intense north-easterly Bora wind event in April 2003 (Figure S3). Numerical simulations were performed either by using the classic formulation (Eq. 1) (black thin line) or computing T considering Eq. 3.*

*Using the classic formulation, the model is not able to correctly compute the time evolution of the local turbidity at near-threshold conditions, especially at the beginning of the event (first peak on the 2ⁿᵈ April 2003). On the contrary, the agreement between measured and computed data is quite good when using the stochastic approach.*

For the Reviewer's convenience, we report here Figure S3 added to the Supplementary information

[Figure]

**Figure S3. Stochastic approach for the computation of suspended sediment transport.** Comparison of measured (dashed line) and computed (solid lines) suspended sediment concentration at the station LT7 (see Figure S2 for the location). The computed suspended sediment concentration was computed by estimating the transport parameter, T , (i) following the probabilistic approach (black bold line, Eq. 3; (ii) using the classic formulation (Eq. 1) without modifying the critical shear stress value (black thin line); (iii) using the classic formulation and reducing the critical shear stress value ($\tau_{cr,S} = 0.3$ Pa; $\tau_{cr,M} = 0.6$ Pa; grey line)

RC2.5: Line 182. In the first paper, the critical shear stress is a well-defined concept. While in Part 2, it is not clear what is the physical meaning of the "minimum-intensity threshold $C_0$" of SSC.
RC2.6: Line 201. The choice of $C_0 = 40$ mg/L needs to be justified.

AR:     These two observations are closely related, so we decided to merge our responses to RC2.5 and 2.6.

We agree with the Reviewer's observations that the choice of the threshold value $C_0$ needs to be explained more in detail. To this aim, following also the suggestions of Reviewer 1, we have completely rewritten the subsection where we introduce the Peak-Over-Threshold analysis as follows:

(line 324) ~~Sediment transport dynamics in tidal environments are the results of the complex interplay among hydrodynamic, biologic, and geomorphologic processes. This interplay between different factors can be fully framed only by taking into account both its deterministic and stochastic components. As an example, Carniello et al. (2011) argued that morphological dynamics in the Venice Lagoon are mostly linked to a few severe resuspension events induced by wind waves, whose dynamics are markedly stochastic in the present configuration (D'Alpaos et al., 2013; Carniello et al., 2016). Measurements confirm that high SSC events are also important sediment suppliers for salt marshes (Tognin et al., 2021).~~

[revised manuscript text omitted]

~~According to the extreme value theory, a Poisson process emerges from a stochastic signal whenever enough high censoring threshold is chosen (Cramér and Leadbetter, 1967). However, as this present analysis is designed to remove only the weak resuspension events induced by periodic tidal currents, the critical threshold is well below the maximum observed values. As a consequence, the aim of the proposed analysis is to characterize the bulk effect of morphologically meaningful SSC events, rather than to describe the extreme events. Notwithstanding the increasing popularity of Poisson processes for the analytical modelling of the long-term evolution of geophysical processes controlled by stochastic drivers in hydrological and geomorphological sciences (e.g., Rodriguez-Iturbe et al., 1987; D'Odorico and Fagherazzi, 2003; Botter et al., 2013; Park et al., 2014; Bertassello et al., 2018), only in the last few years this approach has been adopted for tidal systems (D'Alpaos et al., 2013; Carniello et al., 2016) and the applications portray an encouraging framework.~~

For the Reviewer's convenience, we report here Figure S6 added to the Supplementary information showing the results of the KS test using different $C_0$ thresholds.

[Figure]

**Figure S6. Sensitivity analysis of the threshold $C_0$.** Spatial distribution of Kolmogorov-Smirnov (KS) test at significance level ($\alpha = 0.05$) for different values of the threshold, $C_0$: (a) 30 mg/l; (b) 40 mg/l; (c) 50 mg/l; (d) 60 mg/l. In the maps we can distinguish areas where the KS test is: not verified (dark blue); verified for all the considered stochastic variables (interarrival time, intensity over the threshold and duration) (dark red); verified for the interarrival time and not for intensity and/or duration (light red). Maps show little to no influence of the threshold value within the selected range on the possibility to model over-threshold SSC events as a Poisson process.

RC2.7: Line 343. When talking about the comparison in the past centuries, it might be important to consider the impact of climate change.

AR:  We thank the Reviewer for the comment, which helped us to clarify some of our modelling choices.

As we have already explained in the revision of the companion paper, relative sea level changes surely play a fundamental role in shaping the lagoon morphology and, therefore, we cannot neglect this process. Indeed, we accounted for sea level changes, because each bathymetrical survey and, hence, the correspondent elevation of the computational grid was referred to the coeval mean sea level. Thanks to the Reviewer's comment we realized that this important concept was not properly described in the original version of the manuscript and, therefore, we better highlighted it in the revised version, as follows:

> (line 170) *Each bathymetry and, hence, the elevation of grid elements refers to the local mean sea level at the time when each survey was performed.*
>
> (line 295) *Because bed elevation in each computational grid refers to the mean sea level at the time of each survey, we implicitly take into account the effects of historical relative sea level variations.*

Moreover, we must repeat that, although climate changed in the last four centuries, human interventions undeniably played a primary role in affecting the morphological changes in the same period (Carniello et al., 2009; D'Alpaos, 2010a, 2010b; Finotello et al., 2023; Silvestri et al., 2018). Therefore, we can conclude that the effects of changes in climate on the morphology of the Venice Lagoon are small compared to those resulting from human interventions. In the revised version of the manuscript, we highlighted this concept by modifying the text as follows:

> (line 144) *The Venice Lagoon (Figure 1) underwent different morphological changes over the last four centuries, mainly associated with anthropogenic modifications (Carniello et al., 2009; D'Alpaos, 2010; Finotello et al., 2023).*
>
> (line 162) *As a result, the morphological evolution of the lagoon in the last four centuries has been strongly affected by anthropogenic interventions, along with sea level rise.*

As already noted for the bottom shear stress analysis in the companion paper, we must also stress that, rather than trying to reconstruct the exact climate forcing that gave rise to the present-day morphology, this study aims to understand how morphological changes affect the parameters driving sediment reworking. From this point of view, setting the same boundary conditions is necessary to highlight the specific role of the morphology in affecting the hydrodynamic and wave fields and, therefore, the sediment transport process. By using different boundary conditions for the different historical configurations (although clearly impossible due to data unavailability), it would have been impossible to distinguish the effect on SSC related to the morphology and those due to the different boundary conditions. Indeed, one of the main advantages of numerical modelling is the possibility to isolate the effects of one single parameter, that, in this case, is the morphology, to unravel mechanisms that are otherwise intermingled within a variety of processes.

We have better highlighted this aspect in several points of the revised manuscript:

> (line 13) *The analysis with the peak-over-threshold theory of SSC time series computed using a fully-coupled, bi-dimensional model allow us to identify interarrival times, intensities and durations of over-threshold events and test the*

*hypothesis of modelling SSC dynamics as a Poisson process. The effects of morphological modifications on spatial and temporal SSC patterns are investigated in the Venice Lagoon, for which several historical configurations in the last four centuries are available.* Our results show that, similarly to erosion events, SSC can be modelled as a marked Poisson process in the intertidal flats for all the *different morphological*considered historical configurations *considered*of the Venice Lagoon because exponentially distributed random variables well describe interarrival times, intensity and duration of over-threshold events. Moreover, interarrival times, intensity and duration describing local erosion and over-threshold SSC events are highly related, although not identical because of the non-local dynamics of suspended sediment transport related to advection and dispersion processes. Owing to this statistical characterization of SSC events, it is possible to generate synthetic, yet realistic, time series of SSC for the long-term modelling of shallow tidal environments. *Although erosion and resuspension are intimately intertwined, erosion alone does not suffice to describe also SSC because of the non-local dynamics due to advection and dispersion processes. The statistical characterization of SSC events completes the framework introduced for erosion mechanics and together they represent a promising tool to generate synthetic, yet realistic, time series of shear stress and SSC for the long-term modelling of tidal environments.*

*(line 293) Forcing all the historical configurations of the Venice Lagoon with the same wind and tidal conditions enables us to isolate the effects of the morphological modifications on the wind-wave field, hydrodynamics and sediment dynamics.*

RC2.8: In the introduction, the author stated that there exists field-measured data and remote sensing data for SSC. I think it is necessary to show the comparison between the simulated results and those datasets as model validation.

AR:   We believe that reporting in the main text a detailed description of the model performance against measured data would importantly lengthen the manuscript without a real benefit for the reader because the validation has already been published in more than one paper.
      However, we understand the Reviewer's concern and we deem that the best solution is adding a paragraph on the model performance in the Method section, to provide the reader with more information on the capability of the model to describe the sediment transport dynamics, leaving additional Figures showing a detailed comparison between numerical modelling and measured data in the Supplementary Information. The revised version of the main text now reads:

*(line 273) The model has been widely calibrated and tested in the most recent configuration of the Venice Lagoon, i.e., when field data are available. Since the hydrodynamic model performance has been reported when considering the erosion dynamics (D'Alpaos et al., 2023), here we summarize the ability of the sediment transport model to reproduce SSC by reporting the standard Nash-Sutcliffe Model Efficiency (NSE) parameter computed when field data are available and refer the interested reader to the Supplementary Information (Figures S4 and S5) and the literature (Carniello et al., 2012; Tognin et al., 2022) for a more detailed comparison. Adopting the classification proposed by Allen et al. (2007), the model performance can be rated from excellent to poor (i.e., NSE > 0.65 excellent; 0.5 < NSE < 0.65 very good; 0.2 < NSE < 0.5 good; NSE < 0.2 poor). The STABEM model is very good to excellent in reproducing SSC ($NSE_{mean} = 0.65$, $NSE_{median} = $*

*0.62, NSE_std= 0.17, statistics are derived from calibration reported in Carniello et al., 2012, their Tables 2 and 3, and Tognin et al. (2022), their Table S2). Importantly, the sediment transport model not only correctly reproduces the magnitude of the SSC but also captures its modulation induced by tidal currents and wind-wave variations (Figures S4 and S5).*

A more detailed explanation, along with figures, is reported in the Supplementary Information as follows:

*Supplementary Information*

*Model calibration*

*The calibration of the model was performed for different periods when SSC field measurements are available (Figure S2), but, for the sake of brevity, we report in the following some examples related to intense and weak resuspension events, referring the reader to Carniello et al. (2012) for a more detailed analysis. Model capability to capture the process is evaluated by means of the Nash-Sutcliffe Model Efficiency (NSE) (Allen et al., 2007).*
*Focusing on intense resuspension events, we report in the following the results of three periods characterized by intense wind speed, namely: i) 2–5 April 2003 characterized by Bora wind with speeds up to 16 m/s; ii) 9–13 December 2005 characterized by Bora wind with speeds up to 20 m/s; iii) 29 July–2 August 2007 characterized by Bora wind with speeds up to 20 m/s. The comparison with measured data for simulations with an intense wind event in different stations located within the lagoon shows that the model performance to reproduce SSC is from very good to excellent (NSE_mean = 0.70, NSE_median = 0.67, NSE_std = 0.13). In these inner stations, the main contribution to SSC is provided by the mud fraction being negligible the sand content far from the inlets. Interestingly, the model not only correctly predicts the magnitude of the SSC but also captures its modulation induced by tidal level and wind-wave variations (Figure S4).*
*Additional tests were also carried out considering events characterized by very low wind speed. In particular, we report the results for the period 12-18 April 2006 when turbidity data at six stations close to three inlets are available. At these locations, the sand contribution to the computed SSC is relatively large and of the same order of magnitude as mud contribution (i.e. mud concentration ~10 mg/l and sand concentration ~2–3 mg/l) because close to the inlets the bed composition is richer in sand content than in the inner lagoon. Also in this case the model performance is rated from very good to excellent (NSE_mean = 0.62, NSE_median = 0.59, NSE_std = 0.13). The numerical model correctly reproduces SSC also in the case of very low wind velocity, when the measured SSC can be one order of magnitude lower than that measured with high wind speed (Figure S5).*

For the sake of brevity, we do not report here the Figures we added in the Supplementary Information (see Figure S2 to S5).

RC2.9: In the captions in Figure 2, 3, 4, 5, 7, 8, 9 The description of subfigures is confusing. Recommend switching the order of the year and sub-figure numbering. For example, use "(a) 1611; (b) 1810; ..." instead.

AR: We thank the Reviewer for this suggestion. Done.

---

## Author Response (AR2)

**Author Response to Reviews of Earth Surface Dynamics Manuscript egusphere-2023-320**

**Statistical characterization of erosion and sediment transport mechanics in shallow tidal environments. Part 2: suspended sediment dynamics**

Davide Tognin[1,2], Andrea D'Alpaos[2], Luigi D'Alpaos[1], Andrea Rinaldo[1,3], and Luca Carniello[1]

[1] *Department of Civil, Environmental, and Architectural Engineering, University of Padova, Padova, Italy*
[2] *Department of Geosciences, University of Padova, Padova, Italy*
[3] *Laboratory of Ecohydrology ECHO/IEE/ENAC, Ècole Polytechnique Fèdèrale de Lausanne, Lausanne, Switzerland*
Correspondence: Davide Tognin (davide.tognin@unipd.it)

**Summary**

The authors wish to thank the Editorial Board and the Reviewers for their suggestions. We carefully considered and extensively discussed the possibility of merging the two papers. However, we hold major reservations about merging the two contributions, as we firmly believe that our work can be most effectively conveyed through two separate papers.

As explained more in detail in the following, the main rationale for keeping the two manuscripts separate is content-related, as each paper conveys a distinct message. The overarching contribution of the two companion papers is to test the hypothesis of using random processes to upscale morphodynamic models. However, this cannot be limited to the analysis of erosion events presented in Part 1, because suspended sediment dynamics is not solely influenced by local resuspension but also by advective and mixing processes occurring at the basin scale. Therefore, the characterization of both erosion events and suspended sediment dynamics as Poisson processes is necessary to test the possibility of implementing a synthetic modelling framework accounting for erosion and deposition. This highlights that the two papers are not mere repetitions but rather they address complementary questions on different morphological processes.

To better highlight the complementarity of these works, we have deeply revised the introduction of both papers, as detailed below. Moreover, we have provided a more detailed explanation for the selection of the threshold on suspended sediment concentration, as requested by Reviewer 2.

In the following, we discuss in detail all Reviewers' comments and show how we have addressed them in the revised manuscript, referencing line numbers in the revised manuscript with the track changes.

Please note that the Reviewers' comments are in blue, our detailed responses are in black, and the text of the revised manuscript is framed.

*Legend*

RC:    Reviewer Comment

AR:    Author Response

☐ :    Modified manuscript text

*Note*: References to reviewers' comments are indicated as RCx.x and numbered progressively.

**Reply to Reviewer #1**

RC1.1: I remain very skeptical about the scientific significance of this work with regards of its companion paper (Part 1). The authors admit that the choice of making two papers is (at least partially) driven by the fact that making only one paper would result in a very long paper with too many figures. But I would argue that it's often the case and that requires synthesizing effort to focus the paper on its essential message.

AR: First of all, it is worth mentioning that the decision to keep the two manuscripts separate was not primarily driven by the issue of avoiding an excessively lengthy paper. Instead, it is a practical consideration among various other factors. However, we gave careful thought to the idea of synthesizing the two manuscripts into one single paper. Upon attempting to do so, we realized that too much fundamental material should have been relocated into the Supplementary Information. This is because there are fundamental differences between the two physical processes at hand, namely bottom shear stress (BSS) and suspended sediment concentration (SSC), that deserve to be highlighted to explain the spatio-temporal dynamics of sediment erosion and suspended sediment concentration, which is the key message of our study. The partial overlap is limited to the introduction and the method section, particularly regarding the peak-over-threshold analysis, which must anyhow be partially retained to explain the differences in the analysis of BSS and SSC. We may also finally note that the request to further differentiate the two manuscripts was explicitly suggested in the first revision round. As a result, the lengths of the manuscripts to some extent reflect these adjustments.

Regardless of the manuscript length, the main rationale for maintaining the two manuscripts separated is content-related, as each paper has its own message. The most significant contribution of our study is to test the hypothesis to use random processes to upscale morphodynamics models. When describing morphodynamic changes, both erosive and depositional processes play a fundamental role. Erosion is generally related to the local BSS and deposition to the available SSC. The peak-over-threshold analysis of BSS presented in Part 1 proves that erosion dynamics can be modelled as a Poisson process. However, this offers only a partial perspective, as it does not address the possibility of modelling depositional dynamics as a stochastic process. Indeed, SSC is not solely influenced by local erosion because of advective and dispersive processes occurring at the basin scale, and, hence, must be analyzed independently. Therefore, the novelty of Part 2 lies in demonstrating that spatio-temporal dynamics of SSC can also be modelled as a random process, which is not proved in Part 1.

The characterization of both BSS and SSC as Poisson processes is necessary to test the possibility of implementing a synthetic modelling framework accounting for erosion and deposition. This highlights the difference and the complementarity of the results and clearly demonstrates that Part 2 is not a mere repetition of Part 1 but rather a fundamental component of our research.

To further substantiate this concept, we modified the introduction of Part 1 as follows:

> Manuscript egusphere-2023-319
> (line 60) *A different perspective would be to directly consider the stochasticity of morphodynamic processes. From this point of view, the first step is to test the possibility of setting up a statistically-based framework in order to generate synthetic, yet reliable, time series to model the morphodynamic evolution on long-term time scales and compare possible scenarios in a computationally-effective way through the use of independent Monte Carlo realizations. Although the statistical characterization of the long-term behaviour of several geophysical processes is becoming increasingly popular in hydrology and geomorphology (e.g., Rodriguez-*

*Iturbe et al., 1987; D'Odorico and Fagherazzi, 2003; Botter et al., 2007; Park et al., 2014), applications to tidal landscapes are still quite rare (D'Alpaos et al., 2013; Carniello et al., 2016).*

*The morphological evolution of tidal systems can be described by Exner's equation:*

$$(1-n)\frac{\partial z_b}{\partial t} + \nabla \boldsymbol{q_b} = D - E \qquad\qquad (1)$$

*where $n$ is the bed porosity, $z_b$ is the bed elevation, $\boldsymbol{q_b}$ is the bedload, $D$ and $E$ are the deposition and entrainment rates of sediment, respectively. In mud-dominated tidal systems, sediment is primarily transported in suspension and the bedload is negligible, hence, the bed level changes can be determined by accurately describing erosion and deposition. Erosion, E, is directly influenced by the local bottom shear stress (BSS), which results from the interaction between tidal currents and wind waves in shallow tidal systems (Green and Coco, 2014). Instead, deposition, D, is linked to the suspended sediment concentration (SSC). However, SSC is largely affected by advection and dispersion processes at a larger scale and, therefore cannot be solely determined by local resuspension. Consequently, to effectively model bed-level variations, it is essential to accurately describe both BSS and SSC. This contribution focuses on characterizing BSS, while the analysis of SSC is presented in the companion paper (Tognin et al., 2023).*

In the introduction of Part 2, we added a very brief recall to Exner's equation presented in Part 1 and discussed the differences in the analysis of SSC as follows:

Manuscript egusphere-2023-320

(line 51) *A comprehensive understanding of morphological processes is key to addressing management and restoration strategies for shallow tidal landscapes. The morphodynamic evolution of these systems can be described by Exner's equation:*

$$(1-n)\frac{\partial z_b}{\partial t} + \nabla \boldsymbol{q_b} = D - E \qquad\qquad (1)$$

*where $n$ is the bed porosity, $z_b$ is the bed elevation, $\boldsymbol{q_b}$ is the bedload, $D$ and $E$ are the deposition and entrainment rates of sediment, respectively. Bedload is usually negligible in mud-dominated tidal systems, because sediment transport mainly occurs in suspension, and, thus, the bed level changes are essentially a function of erosion and deposition processes. In order to complete the stochastic framework introduced by D'Alpaos et al. (2023) for the description of erosion events, this study deals with the statistical characterization of suspended sediment concentration (SSC), considered a proxy for depositional processes.*

*Suspended sediment dynamics in shallow tidal systems are influenced by different hydrodynamic and sedimentological factors that vary over a wide range of spatial and temporal scales (Woodroffe, 2002; Masselink et al., 2014). Both tide and waves represent key drivers controlling sediment entrainment and transport in shallow tidal environments (Wang, 2012), with stochastic wave-forced resuspension occasionally increasing by far cyclic tide-driven sediment reworking, especially under storm conditions. Wave resuspension together with tide- and wave-driven sediment transport give rise to advection and dispersion mechanisms leading to basin-wide sediment movement, which largely affect local suspended sediment dynamics (e.g., Nichols and Boon, 1994; Carniello et al., 2011; Green and Coco, 2014). Owing to the complexity of the underlying processes, suspended sediment dynamics in shallow tidal systems is rather entangled and it is not only linked to the local bottom resuspension. Therefore, to effectively describe suspended sediment transport in shallow tidal systems, a dedicated analysis is required.*

*Several numerical models have been developed to describe sediment transport and different techniques have been proposed to upscale the effects on the morphological evolution of tidal systems. For instance, explorative point-based models are extensively used to understand the relative importance of sediment transport processes, because of their simplified parametrization as well as their great conceptual value (Murray, 2007). Furthermore, their reduced computational burden is ideal for investigating trends over long-term time scales. For these reasons, point-based models have been largely adopted, for example, to examine salt-marsh fate under different sea level rise scenarios at the century time scale (D'Alpaos et al., 2011; Fagherazzi et al., 2012). However, point-based models potentially miss spatial dynamics associated with sediment transport and, hence, might fail to represent interactions between different morphological units. More detailed, process-based models can fill this gap and account for sediment fluxes between different points up to the whole basin scale (e.g. Lesser et al., 2004; Carniello et al., 2012). But, because of the explicit description of the short-term interaction between hydrodynamics and sediment transport, the application of process-based models to the long-term time scale is often computationally expensive or even prohibitive.*

**Reply to Reviewer #2**

RC2.1: The paper has been improved after the reviewing process. However, I think more work will be needed in order to publish this paper as a separate paper that uses identical structure and analysis as its companion paper. In my opinion, the most significant contribution of this paper is to introduce the methodology of using random process to upscale the morphodynamics models. This knowledge gap has been filled by its companion paper. Hence it is not necessary to publish a second paper to repeat it.

AR: We appreciate the Reviewer's recognition of our efforts in revising the initial manuscript. After careful consideration of the comment, we realized that, despite our best efforts in the revision, we could not adequately substantiate the need to keep the two contributions separate. The main reason for keeping the two manuscripts separate is that each paper has a distinct message. As the Reviewer aptly pointed out, the most significant contribution of our study is to test the hypothesis to use random processes to upscale morphodynamics models. When describing morphodynamic changes, both erosive and depositional processes play a fundamental role. Erosion is generally related to the local bottom shear stress (BSS) and deposition to the available suspended sediment concentration (SSC). The peak-over-threshold analysis of BSS presented in Part 1 proves that erosion dynamics can be modelled as a Poisson process. However, this offers only a partial picture, as it does not provide any insights into the possibility of modelling depositional dynamics as a stochastic process. Indeed, SSC is not necessarily linearly related to the local BSS (see for example Eq. 9 in the main text) and it is not solely influenced by local factors because of advective and dispersive processes occurring at the basin scale, and, hence, must be analyzed independently. Therefore, the novelty of Part 2 lies in demonstrating that spatio-temporal dynamics of SSC can also be modelled as a random process, a concept not addressed in Part 1.

Characterizing both BSS and SSC as Poisson processes is necessary to test the feasibility of implementing a synthetic modelling framework that accounts for erosion and deposition. This highlights the difference and the complementarity of the results and clearly demonstrates that Part 2 is not a mere repetition of Part 1, but rather a fundamental component of our research. To further substantiate this concept, we modified the introduction of Part 1 as follows:
* * *
Manuscript egusphere-2023-319

(line 60) *A different perspective would be to directly consider the stochasticity of morphodynamic processes. From this point of view, the first step is to test the possibility of setting up a statistically-based framework in order to generate synthetic, yet reliable, time series to model the morphodynamic evolution on long-term time scales and compare possible scenarios in a computationally-effective way through the use of independent Monte Carlo realizations. Although the statistical characterization of the long-term behaviour of several geophysical processes is becoming increasingly popular in hydrology and geomorphology (e.g., Rodriguez-Iturbe et al., 1987; D'Odorico and Fagherazzi, 2003; Botter et al., 2007; Park et al., 2014), applications to tidal landscapes are still quite rare (D'Alpaos et al., 2013; Carniello et al., 2016).*

*The morphological evolution of tidal systems can be described by Exner's equation:*

$$(1-n)\frac{\partial z_b}{\partial t} + \nabla \boldsymbol{q_b} = D - E \qquad (1)$$

*where $n$ is the bed porosity, $z_b$ is the bed elevation, $\boldsymbol{q_b}$ is the bedload, $D$ and $E$ are the deposition and entrainment rates of sediment, respectively. In mud-dominated tidal systems, sediment is primarily transported in suspension and the bedload is negligible, hence, the bed level changes can be determined by accurately describing erosion and deposition. Erosion, E, is directly influenced by the local bottom shear stress (BSS), which results from the interaction between tidal currents and wind*
* * *
*waves in shallow tidal systems (Green and Coco, 2014). Instead, deposition, D, is linked to the suspended sediment concentration (SSC). However, SSC is largely affected by advection and dispersion processes at a larger scale and, therefore cannot be solely determined by local resuspension. Consequently, to effectively model bed-level variations, it is essential to accurately describe both BSS and SSC. This contribution focuses on characterizing BSS, while the analysis of SSC is presented in the companion paper (Tognin et al., 2023).*

In the introduction of Part 2, we added a very brief recall to Exner's equation presented in Part 1 and discussed the differences in the analysis of SSC as follows:

Manuscript egusphere-2023-320

(line 51) *A comprehensive understanding of morphological processes is key to addressing management and restoration strategies for shallow tidal landscapes. The morphodynamic evolution of these systems can be described by Exner's equation:*

$$(1-n)\frac{\partial z_b}{\partial t} + \nabla \boldsymbol{q_b} = D - E \qquad (1)$$

*where $n$ is the bed porosity, $z_b$ is the bed elevation, $\boldsymbol{q_b}$ is the bedload, D and E are the deposition and entrainment rates of sediment, respectively. Bedload is usually negligible in mud-dominated tidal systems, because sediment transport mainly occurs in suspension, and, thus, the bed level changes are essentially a function of erosion and deposition processes. In order to complete the stochastic framework introduced by D'Alpaos et al. (2023) for the description of erosion events, this study deals with the statistical characterization of suspended sediment concentration (SSC), considered a proxy for depositional processes.*

*Suspended sediment dynamics in shallow tidal systems are influenced by different hydrodynamic and sedimentological factors that vary over a wide range of spatial and temporal scales (Woodroffe, 2002; Masselink et al., 2014). Both tide and waves represent key drivers controlling sediment entrainment and transport in shallow tidal environments (Wang, 2012), with stochastic wave-forced resuspension occasionally increasing by far cyclic tide-driven sediment reworking, especially under storm conditions. Wave resuspension together with tide- and wave-driven sediment transport give rise to advection and dispersion mechanisms leading to basin-wide sediment movement, which largely affect local suspended sediment dynamics (e.g., Nichols and Boon, 1994; Carniello et al., 2011; Green and Coco, 2014). Owing to the complexity of the underlying processes, suspended sediment dynamics in shallow tidal systems is rather entangled and it is not only linked to the local bottom resuspension. Therefore, to effectively describe suspended sediment transport in shallow tidal systems, a dedicated analysis is required.*

*Several numerical models have been developed to describe sediment transport and different techniques have been proposed to upscale the effects on the morphological evolution of tidal systems. For instance, explorative point-based models are extensively used to understand the relative importance of sediment transport processes, because of their simplified parametrization as well as their great conceptual value (Murray, 2007). Furthermore, their reduced computational burden is ideal for investigating trends over long-term time scales. For these reasons, point-based models have been largely adopted, for example, to examine salt-marsh fate under different sea level rise scenarios at the century time scale (D'Alpaos et al., 2011; Fagherazzi et al., 2012). However, point-based models potentially miss spatial dynamics associated with sediment transport and, hence, might fail to represent interactions between different morphological units. More*

> *detailed, process-based models can fill this gap and account for sediment fluxes between different points up to the whole basin scale (e.g. Lesser et al., 2004; Carniello et al., 2012). But, because of the explicit description of the short-term interaction between hydrodynamics and sediment transport, the application of process-based models to the long-term time scale is often computationally expensive or even prohibitive.*

RC2.2: Secondly, as the core concepts, author state that the peak-over-threshold theory (POT) can be applied to suspended sediment concentration (SSC). In my opinion, this "threshold SSC" lacks physical meaning. Although this concept looks similar to the critical shear stress, the critical shear stress has a clear definition and is linked to the soil property. The SSC, however, is linked linearly to the shear stress from the entrainment formula, and it is also determined by local flow conditions, wind, wave and so on. There are too many elements that can impact the SSC. The benefit of introducing a threshold SSC concept is not clear, and the definition will not be universal. As a conclusion, the methodology of this paper is no longer new after its companion paper, and the fundamental concept of this paper is not well defined. As a result, I recommend the author put more work and thoughts on this second paper.

AR: The peak-over-threshold (POT) analysis is a statistical method used to analyze a timeseries and, if possible, derive a statistical characterization of overthreshold events. In general, the threshold does not have a direct physical meaning.

As an example, in hydrology, the POT is widely adopted to describe rainfall events, which usually are characterized by a Generalized Pareto distribution, considered the most suitable for modelling extreme events. The threshold for rainfall intensity lacks a physical meaning and it is not universal. Indeed, it is identified in each specific site in order to separate high-magnitude events from the background noise.

From this perspective, the BSS analysis may be considered particularly fortunate because the BSS threshold can be linked to the concept of critical shear stress for erosion. Nevertheless, even in this case, the threshold value is not unique and site-specific, because several factors (such as grain size, cohesion, compaction, bio-stabilization, etc) make it extremely variable both in space and time.

As explained in our reply to RC2.1, to set up the modelling framework describing both erosion and deposition, the same analysis must be applied to both BSS and SSC. However, in the case of SSC, the threshold may not have a strict physical meaning. Still, this does not contradict the assumption of the POT analysis. Similarly to the threshold selection reported here for rainfall intensity, the SSC threshold is selected to isolate the intense events from the baseline concentration available in suspension related to pseudo-deterministic tidal oscillations. For sure, this threshold is not universally applicable and may vary, but the sensitivity analysis outlined in the paper demonstrates that the differences are limited when selected within a reasonable range.

To better clarify these concepts, we modified the text as follows:

> (line 323) In the POT analysis, the threshold value plays a critical role and its choice deserves careful attention. *As already noted for BSS (D'Alpaos et al., 2023), also SSC is locally influenced by many factors, making the threshold non-universal and highly site-specific. In the case of erosion dynamics, the identification of the threshold with the critical shear stress for erosion seems to be relatively straightforward, offering the advantage of preserving a physical meaning related to the process. Instead, when dealing with SSC, the absence of a clear physical threshold mechanism might complicate the identification of the threshold value.*

*Nonetheless, even though a threshold on SSC may lack a physical meaning, the POT analysis can be performed to statistically characterize the bulk effect of morphologically meaningful SSC events.*  *To this aim, the choice of a threshold value, C₀, has to meet two opposite requirements.*  On the one hand, stochastic sediment concentration generated by storm-induced wind waves can be distinguished from pseudo-deterministic, tide-modulated daily concentration only if $C_0$ is large enough. On the other hand, too high values of $C_0$ either require a long, computationally prohibitive simulated time series or can lead to a non-informative analysis because of the large number of events unaccounted for. These observations narrow the range in which the threshold can be selected.